



# Lena River biogeochemistry captured by a 4.5-year high-frequency sampling program

Bennet Juhls[1], Anne Morgenstern[1], Jens Hölemann[2], Antje Eulenburg[1], Birgit Heim[1], Frederieke Miesner[1,3], Hendrik Grotheer[4,5], Gesine Mollenhauer[4,5], Hanno Meyer[1], Ephraim Erkens[1,6], Felica Yara Gehde[1], Sofia Antonova[1], Sergey Chalov[7,8], Maria Tereshina[7], Oxana Erina[7], Evgeniya Fingert[7], Ekaterina Abramova[9], Tina Sanders[10], Liudmila Lebedeva[11], Nikolai Torgovkin[11], Georgii Maksimov[12], Vasily Povazhnyi[13], Rafael Gonçalves-Araujo[14], Urban Wünsch[14], Antonina Chetverova[13,15], Sophie Opfergelt[16], Pier Paul Overduin[1]

[1]Permafrost Research, Alfred Wegener Institute Helmholtz Centre for Polar and Marine Research, Potsdam, 14473, Germany
[2]Physical Oceanography of Polar Seas, Alfred Wegener Institute Helmholtz Centre for Polar and Marine Research, Bremerhaven, 27570, Germany
[3]Department of Geosciences, University of Oslo, Oslo, 0316, Norway
[4]Marine Geochemistry, Alfred Wegener Institute Helmholtz Centre for Polar and Marine Research, Bremerhaven, 27570, Germany
[5]MARUM - Center for Marine Environmental Sciences and Faculty of Geosciences, University of Bremen, Bremen, 28359, Germany
[6]Institute for Geosciences, University of Potsdam, Potsdam, 14476, Germany
[7]Faculty of Geography, Department of Hydrology, Lomonosov Moscow State University, Moscow, 129626, Russia
[8]Institute of Ecology and Environment, Kazan Federal University, Kazan 420097, Russia
[9]Lena Delta Nature Reserve, Tiksi, 678400, Sakha Republic, Russia
[10]Department of Aquatic Nutrient Cycles, Institute of Carbon Cycles, Helmholtz Centre Hereon, Geesthacht, 21502, Germany
[11]Laboratory of Permafrost Groundwater and Geochemistry, Melnikov Permafrost Institute, Russian Academy of Sciences, Yakutsk, 677010, Russia
[12]Laboratory of General Geocryology, Melnikov Permafrost Institute, Russian Academy of Sciences, Yakutsk, 677010, Russia
[13]Otto Schmidt Laboratory for Polar and Marine Research, Arctic and Antarctic Research Institute, St. Petersburg, 199397, Russia
[14]Section for Oceans and Arctic, National Institute of Aquatic Resources, Technical University of Denmark, Lyngby, 2800, Denmark
[15]Institute of Earth Science, St. Petersburg University, St. Petersburg, 199034, Russia
[16]Earth and Life Institute, Université catholique de Louvain, Louvain-la-Neuve, 1348, Belgium

*Correspondence to*: Bennet Juhls (bennet.juhls@awi.de)

**Abstract.** The Siberian Arctic is warming rapidly, causing permafrost to thaw and altering the biogeochemistry of aquatic environments, with cascading effects on the coastal and shelf ecosystems of the Arctic Ocean. The Lena River, one of the largest Arctic rivers, drains a catchment dominated by permafrost. Baseline discharge biogeochemistry data is necessary to understand present and future changes in land-to-ocean fluxes. Here, we present a high-frequency, 4.5-year-long dataset from a sampling program of the Lena River's biogeochemistry, spanning April 2018 to August 2022. The dataset comprises 587 sampling events and measurements of various parameters, including water temperature, electrical conductivity, stable oxygen and hydrogen isotopes, dissolved organic carbon concentration and $^{14}$C, coloured and fluorescent dissolved organic matter,



dissolved inorganic and total nutrients, and dissolved elemental and ion concentrations. Sampling consistency and continuity and data quality were ensured through simple sampling protocols, real-time communication, and collaboration with local and international partners. The data is available as a collection of datasets separated by parameter groups and periods at https://doi.org/10.1594/PANGAEA.913197 (Juhls et al., 2020b). To our knowledge, this dataset provides an unprecedented

temporal resolution of an Arctic river's biogeochemistry. This makes it a unique baseline on which future environmental changes, including changes in river hydrology, at temporal scales from precipitation event to seasonal to interannual, can be detected.

## 1 Introduction

River-borne organic material and nutrients influence biogeochemical processes in Arctic estuaries, coastal waters, shelf seas

and, on a larger scale, the entire Arctic Ocean. The Arctic is warming nearly four times faster than the rest of the world (Rantanen et al., 2022), changing ecosystems, the intensity of geomorphic processes and aquatic biogeochemistry within river catchments. These changes are reflected in the flux of matter borne by rivers to the sea. For example, as air temperatures rise, permafrost thaws (Biskaborn et al., 2019), mobilizing and releasing organic matter and nutrients into the aquatic system(Mann et al., 2022; Vonk et al., 2019), changing hydrological pathways (Rawlins and Karmalkar, 2024) and the nature of the organic

material transported (Starr et al., 2024). There is no paleo-historical analogue for these changes, therefore, establishing a baseline of current fluxes and understanding how the system is changing are necessary to anticipate the scope and consequence of future impacts of climate warming and permafrost thaw. The Lena River is the second largest Arctic river by total annual discharge, and its catchment is one of the most rapidly changing in the Arctic (Tananaev and Lotsari, 2022). The consequences of rapid warming are evident in the Lena River's hydrology. Total annual discharge is increasing (Shiklomanov et al., 2020;

Tananaev et al., 2016) and the hydrological regime is shifting towards earlier freshets and later freeze-ups (Gelfan et al., 2017; Yang et al., 2002). The seasonal variability of water sources supplying the Lena River is also changing as a result. For example, winter under-ice flow has been increasing for almost the entire the past century (Liu et al., 2022). Such changes inevitably affect the river's biogeochemistry (Juhls et al., 2020a). The timings of river ice melt and freeze-up are undergoing significant alterations (Shiklomanov and Lammers, 2014), shifting water-atmosphere heat and mass transfer as the ice-free season

lengthens. These changes are reflections of synoptic shifts in climate, which also affect the catchment and the ecosystem function of the river. Shifts in the Lena River biogeochemistry will lead to further changes in the region's climate dynamics and to currently unknown impacts on coastal ecosystems. The Lena River plays a crucial role in the global carbon cycle, transporting large amounts of organic matter from the terrestrial environment to the Arctic Ocean (e.g., Raymond et al., 2007; Semiletov et al., 2011). The Lena River also transports nutrients to shallow shelf and coastal regions, where they are important

for the primary production of associated ecosystems (Terhaar et al., 2021). Riverine dissolved inorganic carbon (DIC) and dissolved organic carbon (DOC) fluxes drive outgassing of greenhouse gasses in the river plume (Bertin et al., 2023), providing one example of a feedback mechanism between changing riverine fluxes and the climate system. Rivers act not only as





conveyor belts, but also transform the material they transport and represent important habitats. Food and transportation security are two of the important ecosystem services the Lena River provides to northern communities. To understand the changes underway, their impacts on the river system and, in turn, their impacts on the global climate, a baseline of observations that includes biogeochemistry is required. It is a prerequisite for deriving improved insights into linkages between land and ocean and between river system and climate that will allow for better constraining future impacts of continued warming. The majority of studies that investigate recent Arctic fluvial biogeochemistry trends (Holmes et al., 2012; Raymond et al., 2007; Tank et al., 2023; Wild et al., 2019) are based on the series of pan-Arctic River sampling programs PARTNERS (2003–2007), Student Partners (2004-2009) and ArcticGRO (since 2009). They systematically cover the six largest Arctic rivers in respect to their discharge, including the Lena River. These programs have produced data over more than two decades, providing ~7 samples per year from each river to cover seasonal changes. Other studies investigate Lena River biogeochemistry with datasets from distinct field campaigns at specific locations or along transects (Cauwet and Sidorov, 1996; Hölemann et al., 2005) and its transport of sediment (Fedorova et al., 2015; Ogneva et al., 2023; Rachold et al., 1996), carbon (Juhls et al., 2020a; Kutscher et al., 2017; Winterfeld et al., 2015) and nutrients (Lara et al., 1998; Sanders et al., 2022). Most of these studies focus on a selected set of parameters for specific research questions and include only the summer or the open water period. Especially the shoulder season with the freshet in spring and the freeze-up in fall are mostly unstudied. Poor temporal resolution and coverage of sampling had to be bridged with models that relate discharge with biogeochemical concentrations (e.g., Holmes et al., 2012; Raymond et al., 2007; Tank et al., 2023). The necessity to use these relationships, which are often weak, can be obviated through higher frequency sampling. Juhls et al., (2020a) compare the effect of calculating annual fluxes using data sets of varying sampling frequency. Higher sampling frequency can improve annual flux estimates, as does dedicated sampling over the whole hydrological cycle. Arctic rivers are typically characterized by a nival hydrological regime, and, thus, the strong seasonality and high variability in summer water balance may mandate high-frequency data collection, especially during the highly dynamic shoulder seasons (freshet, freeze-up). Even more importantly, the assumption of correlations between biogeochemical parameters and river discharge may mask emerging catchment or river processes that are not tied to discharge. A relevant example of this are shifts in hydrologic pathways due to climate change and permafrost thaw (Prokushkin et al., 2019), which may affect organic matter (OM) quality, but not discharge (Frey and Smith, 2005). In addition, higher frequency or even continuous in situ measurements (e.g., Castro-Morales et al., 2022) will create new opportunities to validate remotely sensed data (El Kassar et al., 2023) or model results (e.g. Rawlins and Karmalkar, 2024) and to potentially upscale data spatially. The biogeochemistry of a river is impacted by environmental processes of its entire upstream catchment and may therefore reflect changes across a range of scales (Holmes et al., 2012). In order to record future changes in the Lena River biogeochemistry that are related to climate warming, it is crucial to compare new data with a baseline. In this study, we present biogeochemical data collected from water sampled in the central Lena River Delta over more than four years, along with detailed descriptions of the sampling, processing and analytical methods for each parameter.

## 2 Study area, climatological and hydrological conditions

The Lena River stretches from the Baikal Mountains to the Laptev Sea where it forms the largest Arctic Delta. The Lena River has a total length of about 4,294 km and an average annual discharge of 689.1 km$^3$ year$^{-1}$ (Mann et al., 2022). More than 90 % of its catchment (2.61 x 10$^6$ km$^2$) is underlain by continuous or discontinuous permafrost (Obu et al., 2019). The two major tributaries to the Lena are the Viluy River, from the west, and the Aldan River, from the east (Fig. 1a).

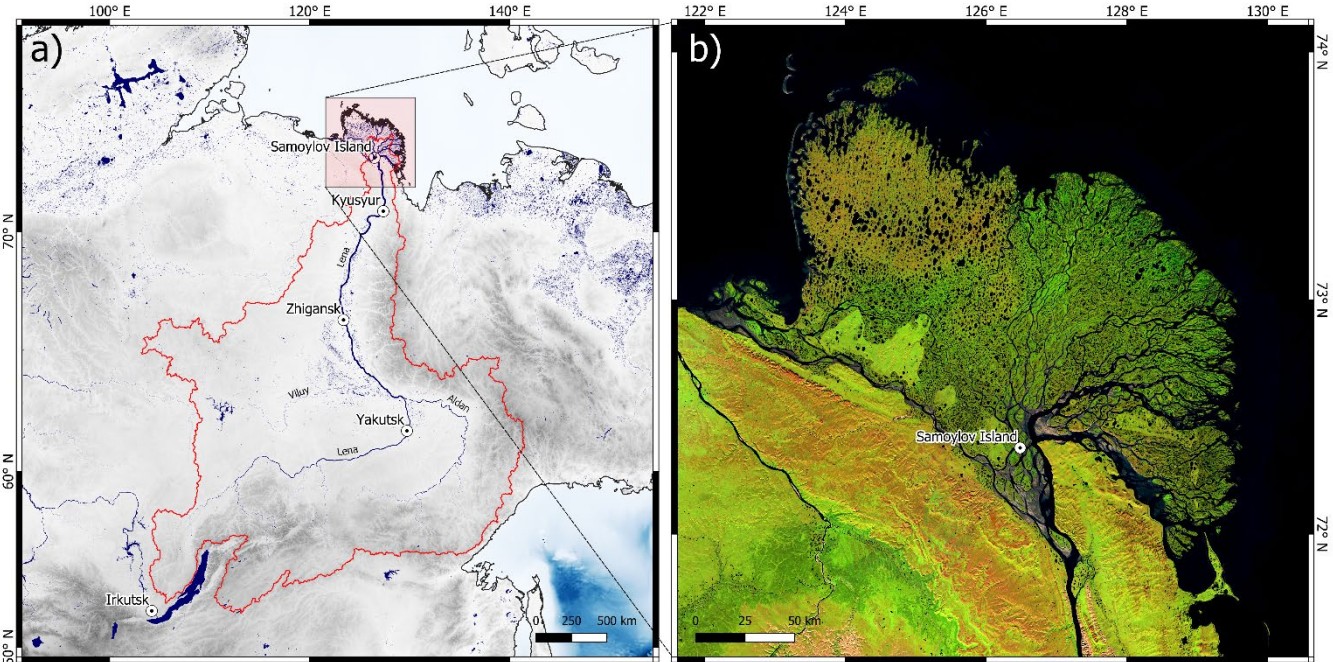

**Figure 1: (a) Overview of the Lena River watershed, which is delineated by red line. Grey colours show topography, blue colours rivers, lakes and seas; (b) satellite image of the Lena River Delta with the sampling location at Samoylov Island.**

The Lena River catchment is one of the coldest regions on Earth with average air temperatures below -30 °C from December to February in most years. Only for five months in the year (May to September) mean air temperatures are above 0 °C. The catchment climate is characterized by a dry winter, and most of the annual precipitation occurs during the summer months (Fig. 2). During the period covered by the sampling program in this study, the mean monthly air temperature was mostly above the long-term average (1950-2022). The precipitation for these years was mostly higher than the long-term average (1950-2022) during the winter months, but very variable during the summer months, including record low (in August 2020 and October 2022) and record high (in March 2020 and August 2022) monthly precipitation.

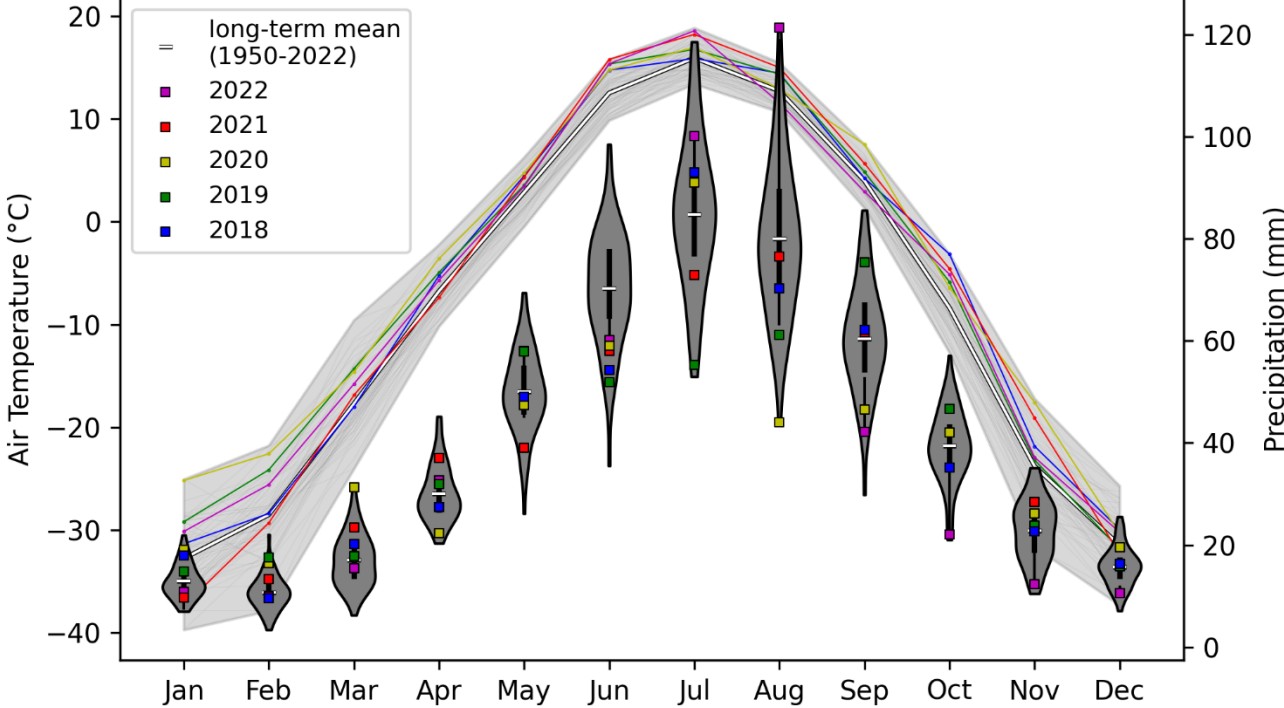

**Figure 2: ERA5-land monthly mean air temperature and total precipitation for the Lena River catchment. Lines show the air temperatures, where the light gray area indicates the min-max range of all years from 1950 to 2022. Precipitation is shown by violins, where the width indicates the occurrence frequency within the years from 1950 to 2022. The years 2018 to 2022 are highlighted by colored lines (temperature) and squares in violins (precipitation). Data source: ERA5. Credit: Copernicus Climate Change**
**Service/ECMWF**

The Lena River is characterized by a nival hydrological regime with a strong discharge peak during the snowmelt and river ice break-up between the end of May and the beginning of June, a variable discharge in summer, and low base flow discharge in winter (Fig. 3, 4a). Daily discharge is monitored by the Russian Federal Service for Hydrometeorology and Environmental Monitoring (Roshydromet). All 4 years covered by the sampling program described in this study showed higher than average
winter discharge, but ranged from record low to record high summer discharge. While 2018 was the year with the fourth highest annual discharge on record (1936 to 2022), 2019 was one of the driest years (ninth on record).



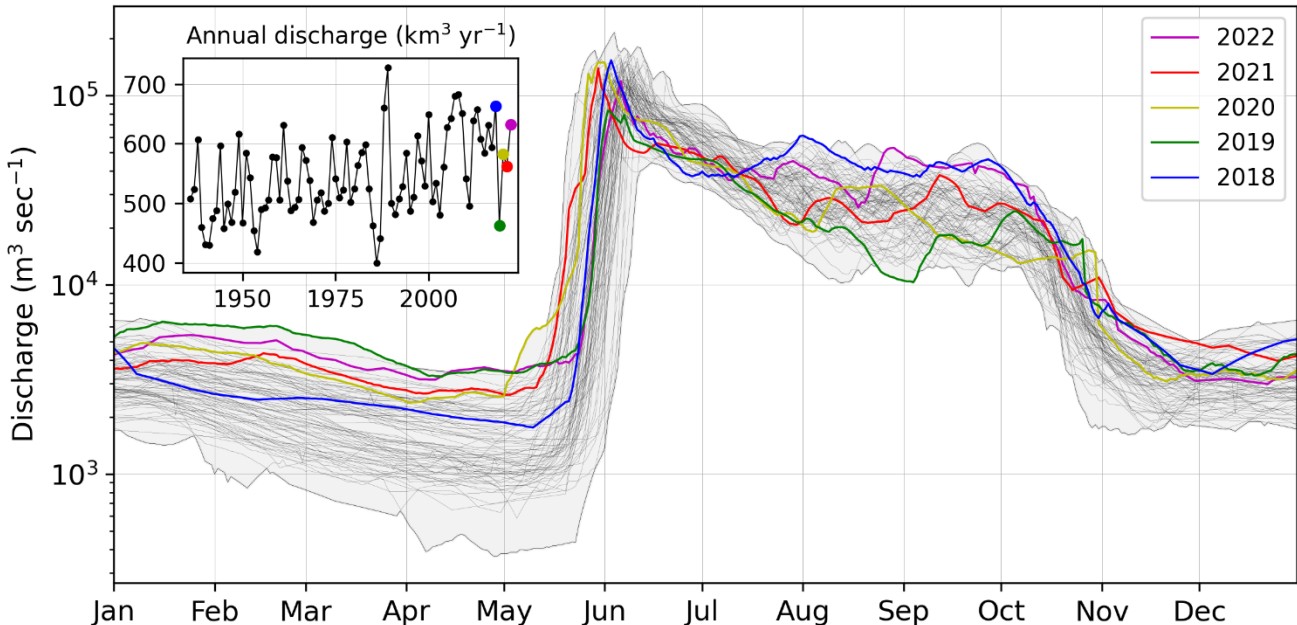

**Figure 3: Discharge of the Lena River for all years from 1937 to 2022 (thin black lines) measured at Kyusyur. The years 2018 to 2022 are highlighted in colours. Grey area shows the absolute minimum and maximum for each day from 1937 to 2022. Inset figure (top left) shows the annual discharge fluxes.**

## 3 Data Collection

The dataset is derived from a high-frequency sampling program at the Research Station Samoylov Island in the central Lena River Delta (Fig. 1b), a permanently staffed research station since 2013. It relied on the support of the non-scientific station staff and was thus designed as robust and time efficient under all seasonal conditions as possible. The exact location of sampling was the Olenekskaya Channel, south-west of Samoylov Island. The sampling location is also directly connected to the near main Lena River channel (~7 km upstream). Only surface samples were taken, and most samples were taken from the center of the channel from a boat (in summer) or through an ice hole (in winter). No stratification can be observed in the Lena River (Fig. A1), suggesting homogeneously distributed dissolved water chemistry across the water column. During ice break-up and ice freeze-up, samples were taken from the shore for safety reasons during unstable river ice conditions. The sampling started on 20 April 2018 and lasted until 16 August 2022. Throughout the first year of sampling (18 April 2018 to 6 April 2019), a sample was taken every 4 days. Between 10 May to 14 June each following year, a sample was taken every day, between 15 June to 31 October every 2 days, and between 1 November to 9 May once a week. For each sampling event, 1 L of surface water was taken in a Nalgene plastic bottle pre-rinsed with river water, and the temperature was measured directly in the river using a hand-held conductivity meter. The water sample was then immediately subsampled, filtered and conserved in the laboratory of the research station. Additionally, time of sampling, exact location (center of channel or shore), and weather



and river conditions were noted. Except for the in situ temperature measurement, all biogeochemical analyses were done after storage and transport to laboratories in Russia, Germany, and Denmark. As a result, some samples were stored up to half a year before analysis. After initial processing, samples were stored in refrigerators (+4°C) and freezers (-18°C) at the Research Station Samoylov Island until transport via Tiksi to Yakutsk, from where they were shipped to the individual labs. Dates

separating sample sets (with distinct transport and analysis dates) are indicated by the dashed lines in the individual figures for each parameter. The dataset contains data from 587 sampling dates. This sampling program focuses on biogeochemical parameters, which allow a simple processing protocol to be sampled and processed by a non-scientist after initial training. It includes biogeochemical parameters mostly in the dissolved phase and excludes parameters measured on particles, because in contrast to dissolved concentrations the river channel at Samoylov Island might not be assumed representative for the main

Lena River for particulates concentrations due to their strong heterogeneity (Chalov and Prokopeva, 2021). During the first days of the spring freshet, which are accompanied by ice jams and pooling of snow and ice melt water, representative sampling of Lena River water is challenging if not impossible. Consequently, the samples during these periods can show biogeochemical signatures of dilution and need to be interpreted with caution. The periods of ice cover on the Lena River were visually estimated using daily MODIS imagery (https://worldview.earthdata.nasa.gov/) and shown by the white (ice-covered) and grey

(ice-free) backgrounds in the individual figures for each parameter. Table 1 provides an overview of the sample processing and analysis methodology for each biogeochemical parameter and each set of samples.

**Table 1: Overview of sample processing and analysis for each group of parameters for each set of samples.**

| Parameter | Period / Sample ID | Sample processing | Method, Instrument, Location | Accuracy/Error |
|---|---|---|---|---|
| Temperature | 20 Apr 2018 (#001) to 16 Aug 2022 (#612) | in situ measured | Handheld conductivity meter (WTW COND 340i), in situ | ±0.5 % |
| Electrical conductivity (EC) | 20 Apr 2018 (#001) to 6 Apr 2019 (#078) | unfiltered & cooled | Conductivity meter (WTW Multilab 540), Alfred Wegener Institute in Potsdam, Germany (AWI) | ±0.5 % |
| | 11 Apr 2019 (#079) to 11 Sept 2019 (#201) | unfiltered & frozen, thawed before analysis | | |



| Parameter | Period | Treatment | Instrument | Uncertainty |
|---|---|---|---|---|
| | 13 Sept 2019 (#202) to 28 Aug 2020 (#362) | | | |
| | 30 Aug 2020 (#363) to 23 Aug 2021 (#487) | | | |
| | 26 Aug 2021 (#488) to 16 Aug 2022 (#612) | filtered & cooled | Conductivity meter (Milwaukee EC59 PRO), Lomonosov Moscow State University, Russia (MSU) | ±2 % |
| Stable isotopes ($\delta^{18}$O, $\delta$D) | 20 Apr 2018 (#001) to 13 Sept 2018 (#039) | unfiltered & cooled | Finnigan MAT Delta-S mass spectrometer, Alfred Wegener Institute in Potsdam, Germany (AWI) | $\delta$D: ±0.8 %, $\delta^{18}$O: ±0.1 % |
| | 29 Sept 2018 (#043) to 6 Apr 2019 (#078) | | | |
| | 11 Apr 2019 (#079) to 11 Sept 2019 (#201) | | | |
| | 13 Sept 2019 (#202) to 28 Aug 2020 (#362) | | | |





| | 30 Aug 2020 (#363) to 23 Aug 2021 (#487) | | | |
| --- | --- | --- | --- | --- |
| | 26 Aug 2021 (#488) to 02 Aug 2022 (#605) | unfiltered & cooled | PICARRO L2140i CRDS, Melnikov Permafrost Institute in Yakutsk, Russia (MPI) | δD: ±0.8 %, δ$^{18}$O: ±0.1 % |
| Dissolved nutrients (Si, PO$_4$, NH$_4$, NO$_2$, NO$_3$) | 20 Apr 2018 (#001) to 13 Sept 2019 (#39) | filtered & frozen | Continuous flow auto analyzer (San++, SKALAR) Otto Schmidt Laboratory in St. Petersburg, Russia (OSL) | N/A |
| | 17 Sept 2018 (#040) to 11 Sept 2019 (#201) | | | |
| | 13 Sept 2019 (#202) to 23 Aug 2021 (#487) | filtered & frozen | Continuous flow auto analyzer (AA3, SEAL Analytics) Helmholtz-Zentrum Hereon in Geesthacht, Germany (Hereon) | Detection limits: nitrate: 0.049µM, nitrite: 0.015µM ammonium: 0.092µM Silicate: 0.324µM phosphate: 0.011 µM |
| Total dissolved nitrogen and total dissolved phosphorus (TDN, TDP) & Total nitrogen and total phosphorus (TN, TP) | 20 Apr 2018 (#001) to 13 Sept 2018 (#039) | unfiltered & frozen | Persulfate oxidation and continuous flow auto analyzer (San++, SKALAR) Otto Schmidt Laboratory in St. Petersburg, Russia (OSL) | N/A |
| | 11 Sept 2019 (#201) to 28 Aug 2020 | filtered & frozen | Persulfate oxidation and continuous flow auto analyzer (AA3, SEAL Analytics) | N/A |





| | | | | |
|---|---|---|---|---|
| (#362), only TD/TN | | Helmholtz-Zentrum Hereon in Geesthacht, Germany (Hereon) | | |
| | 26 Aug 2021 (#488) to 16 Aug 2022 (#612) | filtered & frozen | Persulfate oxidation, photometric (PE-5400UV spectrophotometer by ECROS LLC), Lomonosov Moscow State University in Moscow, Russia (MSU) | Standard error at p=0.05: TP/TDP: 0.0001+0.08*mg P/L TN/TDN: 0.04+0.077*mg N/L |
| DOC | 20 Apr 2018 (#001) to 13 Sept 2018 (#039) | filtered & acidified & cooled | High temperature catalytic oxidation, TOC-VCPH (Shimadzu), Alfred Wegener Institute in Potsdam, Germany (AWI) | <5 % (against 8 standards of different concentrations) |
| | 29 Sept 2018 (#043) to 6 Apr 2019 (#078) | | | |
| | 11 Apr 2019 (#079) to 11 Sept 2019 (#201) | | | |
| | 11 Sept 2019 (#201) to 2 May 2020 (#287) | | | |
| | 9 May 2020 (#288) to 28 Aug 2020 (#362) | | | |





| | 30 Aug 2020 (#363) to 23 Aug 2021 (#487) | | | |
|---|---|---|---|---|
| | 26 Aug 2021 (#488) to 16 Aug 2022 (#612) | filtered & acidified & cooled | High temperature catalytic oxidation, TOPAZ NC, Informanalitika LLC, Lomonosov Moscow State University in Moscow, Russia (MSU) | <5 % (against 4 standards of different concentrations) |
| $a_{CDOM}(\lambda=200\text{-}800$ nm) | 20 Apr 2018 (#001) to 13 Sept 2018 (#039) | filtered & cooled | Spectrophotometer, SPECORD 200 (Analytik Jena), Otto Schmidt Laboratory in St. Petersburg, Russia (OSL) | 1 cm or 5 cm cuvette, spectral resolution: 1.6-1.8 nm |
| | 29 Sept 2018 (#043) to 6 Apr 2019 (#078) | filtered & cooled | Spectrophotometer, LAMBDA 950 UV/Vis (PerkinElmer), German Research Centre for Geosciences in Potsdam, Germany (GFZ) | 1 cm or 5 cm cuvette, spectral resolution: <0.05 nm |
| | 11 Apr 2019 (#079) to 11 Sept 2019 (#201) | filtered & cooled | Spectrophotometer, SPECORD 200 (Analytik Jena), Otto Schmidt Laboratory in St. Petersburg, Russia (OSL) | 1 cm or 5 cm cuvette, spectral resolution: 1.6-1.8 nm |
| | 13 Sept 2019 (#202) to 23 Aug 2021 (#487) | filtered & cooled | Spectrophotometer, LAMBDA 950 UV/Vis (PerkinElmer), German Research Centre for Geosciencesin Potsdam, Germany (GFZ) | 1 cm or 5 cm cuvette, spectral resolution: <0.05 nm |



| Parameter | Period | Storage | Instrument | Notes |
|---|---|---|---|---|
| | 26 Aug 2021 (#488) to 16 Aug 2022 (#612) | filtered & cooled | Spectrophotometer, PE-5400-UV (ECROS LLC), Lomonosov Moscow State University in Moscow, Russia (MSU) | 2 cm cuvette |
| FDOM | 20 Apr 2018 (#001) to 13 Sept 2018 (#039) | filtered & cooled | Horiba AquaLog, Technical University of Denmark in Lyngby, Denmark (DTU) | 1 cm cuvette |
| | 29 Sept 2018 (#043) to 6 Apr 2019 (#078) | | | |
| | 11 Apr 2019 (#079) to 11 Sept 2019 (#201) | | | |
| | 13 Sept 2019 (#202) to 23 Aug 2021 (#487) | | | |
| DOC radiocarbon | 30 Sept 2019 to 15 July 2021 | frozen | EA-GIS-MICADAS, Alfred Wegener Institute in Bremerhaven, Germany (AWI) | 2σ mean = 18 ‰ |
| Ions ($SO_4$, Cl, Br, F, $NO_3$, $PO_4$) | 20 Apr 2018 (#001) to 28 Mar 2019 (#077) | unfiltered & cooled, filtered before analysis | Ion chromatography, ICS 2100 (Thermo-Fischer), Alfred Wegener Institute in Potsdam, Germany (AWI) | Detection limits: F, Br = 0.05 mg L$^{-1}$, Cl, $SO_4$, $NO_3$ = 0.1 mg L$^{-1}$ |



|  |  |  |  |  |
|---|---|---|---|---|
|  | 6 Apr 2019 (#078) to 11 Sept 2019 (#201) | unfiltered & frozen, thawed & filtered before analysis |  |  |
|  | 13 Sept 2019 (#202) to 2 May 2020 (#287) |  |  |  |
|  | 9 May 2020 (#288) to 28 Aug 2020 (#362) |  |  |  |
|  | 30 Aug 2020 (#363) to 23 Aug 2021 (#487) |  |  |  |
| Ions (Cl, SO₄, F, NO₃) | 26 Aug 2021 (#488) to 16 Aug 2022 (#612) | unfiltered & frozen, thawed & filtered before analysis | Ion chromatography, Concise ICSep An2, Lomonosov Moscow State University in Moscow, Russia (MSU) | N/A |
| Total dissolved elemental concentration (Al, Ba, Ca, Fe, K, Mg, Mn, Na, P, Si, Sr) | 20 Apr 2018 (#001) to 28 Mar 2019 (#077) | unfiltered & cooled, filtered & acidified before analysis | Inductively coupled plasma optical emission spectroscopy, ICP-OES Optima 8300DV (Perkin Elmer), Alfred Wegener Institute in Potsdam, Germany (AWI) | Detection limits: Al, Ba, Fe, K, Na, Sr = 0.2 mg L⁻¹, Ca, Mg, Mn, P, Si = 0.1 mg L⁻¹ |
|  | 6 Apr 2018 (#078) to 2 May 2020 (#287) | unfiltered & frozen, thawed & filtered & acidified before analysis |  |  |
|  | 9 May 2020 (#288) to 28 |  |  |  |





| | Aug 2020 (#362) | | | |
| --- | --- | --- | --- | --- |
| Total dissolved elemental concentration (Na, K, Mg, Ca, Si, NH$_4$) | 26 Aug 2021 (#488) to 16 Aug 2022 (#612) | unfiltered & frozen, thawed & filtered & acidified before analysis | Ion chromatography, Shodex IC YS-50, Lomonosov Moscow State University in Moscow, Russia (MSU) | N/A |

## 3.1 River water temperature and electrical conductivity

For all sampling events (#001 to #612), the temperature of the river water was measured during sampling directly in the river
at about 20 cm water depth using a handheld WTW COND 340I (accuracy ±0.5 %). The electrical conductivity (EC) of samples #079 to #487 (6 April 2019 to 23 August 2021) was measured on frozen samples that were thawed (24h at room temperature) after transport to the hydrochemistry laboratory at the AWI in Potsdam, Germany, using a WTW Multilab 540 conductivity meter (accuracy ±0.5 %). The EC of the first year (20 April 2018 to 6 April 2019; sample #001 to #078) were measured on unfrozen samples. Before each series of EC measurements, the conductivity meter was calibrated (cell constant was set for
25°C reference temperature) with a standard-solution with 1413 µS cm$^{-1}$. Between samples, the conductivity meter was cleaned with Milli-Q water and wiped dry. The EC for the last year (#488 to #611, 6 August 2021 to 14 August 2022) was measured on filtered samples (0.45 µm cellulose acetate) at the Lomonosov Moscow State University in Moscow (MSU) using a Milwaukee EC59 PRO conductivity meter (accuracy ±2 %). The river water temperature ranged between -0.3 and 20.2°C (Fig. 4b). During the ice-covered period, the river water temperature was very stable between -0.3 and 0.2°C. After ice break-up,
the river water temperature increased from ~0°C to above 15°C within about two weeks. Between August and September, the river water temperature started to drop until reaching 0°C in mid-October.





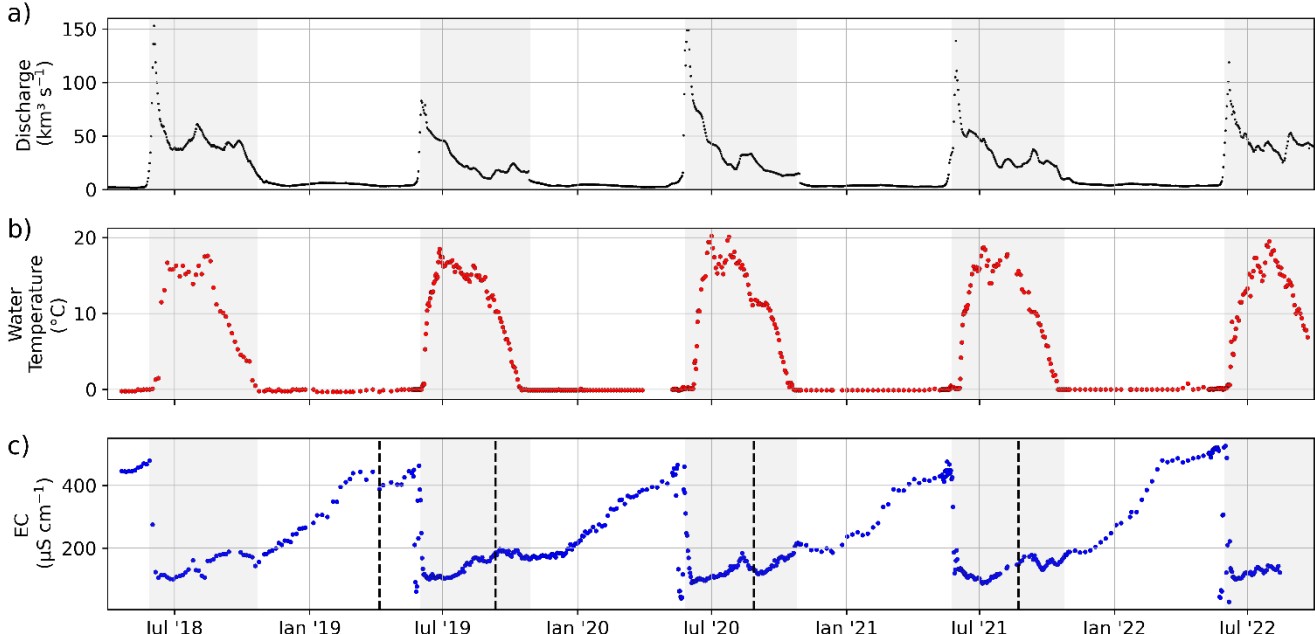

**Figure 4: Time series of (a) discharge measured at Kyusyur and (b) river water temperature and (c) EC. Gray areas indicate the ice-free periods. Dashed black lines separate sample sets and indicate a change in measurement protocol and method (see Table 1).**

The EC ranged between 29 and 526 µS cm$^{-1}$ (Fig. 4c). Highest EC were observed at the end of the winter, right before ice break-up and the freshet of the river. In 2018, 2019, 2020, and 2022, the peak of the spring flood coincided with the lowest annual EC. Only in 2021, the EC was lower in the first days of July compared to right after ice break-up in the beginning of June. In late winter 2022, EC was higher compared to previous years, whereas the lowest EC during the freshet was on a similar level as in previous years. EC was measured in five series (separated by dashed black lines in Fig. 4c). The data shows no offsets between those series. Note that a small number of samples exhibiting exceptionally low EC levels before the spring freshet could potentially be influenced by the pooling of meltwater from snow and ice during ice jamming events. For the samples #202 to #287 (13 September 2019 to 2 May 2020), we compared the EC from two sets of samples that were 1) frozen right after sampling and filtered only after transport and thaw before analysis, and 2) filtered right after sampling and transported unfrozen (cooled at +4°C) (Fig. B1).

**3.2 Stable isotopes of water**

Water samples for stable isotopes were filled immediately after sampling, untreated, into 10 mL HDPE vials, sealed tightly, and stored in the dark at 4°C. After transport, measurements for samples #001 to #487 (20 April 2018 to 23 August 2021) were conducted in five sample series (see Table 1) at the ISOLAB stable isotope facility at Alfred Wegener Institute in Potsdam (AWI) using a Finnigan MAT Delta-S mass spectrometer equipped with equilibration units for the online determination of hydrogen and oxygen isotopic composition. The measurement accuracy for hydrogen and oxygen isotopes was better than

±0.8 % and ±0.1 %, respectively (Meyer et al., 2000). Analysis for samples #488 to #605 (26 August 2021 to 2 August 2022) were conducted at the Melnikov Permafrost Institute in Yakutsk (MPI) using a PICARRO L2140i Isotopic Water Liquid Analyzer for the online determination of the hydrogen and oxygen isotopic composition in water samples using Cavity Ring-Down Spectroscopy (CRDS). PICARRO L2130i CRDS uses a laser with an effective path length of up to 20 kilometers to

quantify spectral features of gas phase molecules by scanning repeatedly the absorption lines of $H_2^{16}O$, $H_2^{18}O$, and $HD^{16}O$ in a temperature and pressure controlled optical cavity. The PICARRO L2140i Isotopic Water Liquid Analyzer simultaneously measures isotopic ratios of D/H and $^{18}O/^{16}O$ in liquid water providing both, $\delta^{18}O$ and $\delta D$ data from one aliquot. Samples from 2 mL glass vials are automatically injected into a temperature controlled and stabilized Vaporizer Unit (A0211) held at high temperature with the vapor sent to the analyzer. At least three standards are used for quality control, selected according to the

expected isotopic composition of the samples. For a single CRDS stable isotope measurement, about 2 µL of water is injected. This process is repeated six times resulting in both, final dD and $d^{18}O$ values (which refers to the permille difference related to V-SMOW). These values are corrected for drift and memory effects. The precision of long-term standard measurements for the H and O isotope composition is better than ±0.8 ‰ and ±0.10 ‰, respectively. The data are reported as $\delta D$ and $\delta^{18}O$ values, which is the per mille (‰) difference from standard V-SMOW. The deuterium excess (d-excess) is calculated by

$$d - excess = \delta D - 8 * \delta 18O, \tag{1}$$

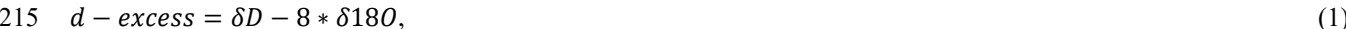
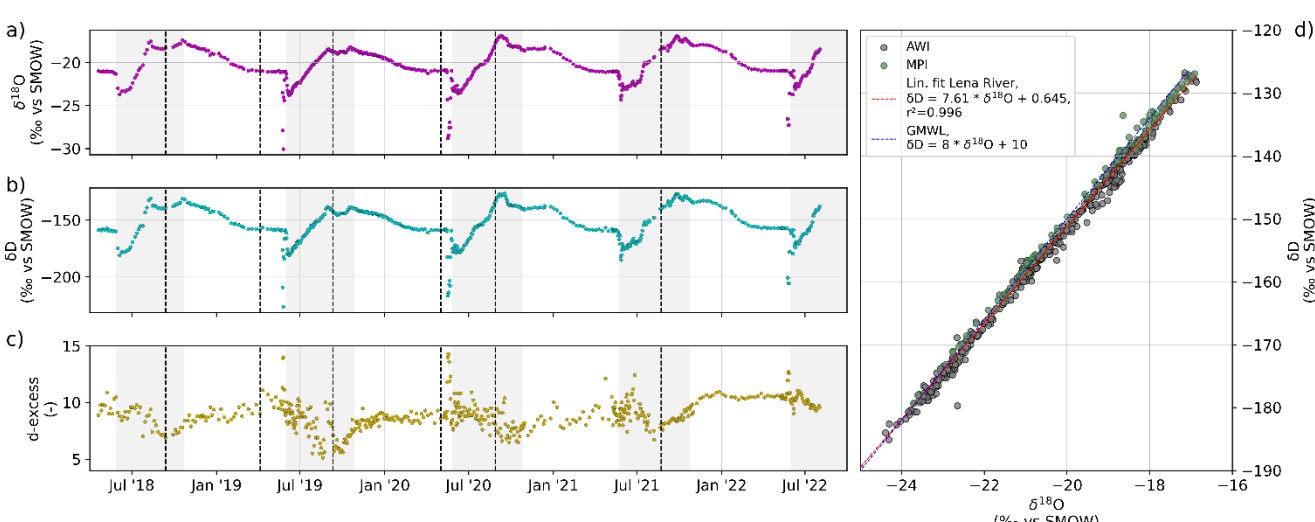

**Figure 5: Time series of (a) $\delta^{18}O$, (b) $\delta D$, and (c) d-excess. The scatterplot in (d) shows the relationship between $\delta^{18}O$ and $\delta D$ of the Lena River, where samples measured at the Alfred Wegener Institute are indicated by a black outline and samples measured at the Melnikov Permafrost Institute by a green outline. The linear regression is shown by the solid red line and the dashed blue line shows**
**the global meteoric water line after Craig (1961).**

Both, $\delta D$ and $\delta^{18}O$ were at their lowest during the beginning of the ice-free season and increased over its course. Late summer and early fall variability due to rainfall events precedes the formation of ice. Once the ice cover started to form, the isotope signature decreased gradually over the fall and winter until dilution with snow melt at break-up restarted the annual cycle. Similar as for EC, a small number of samples during initial breakup had much lower values (< -25 ‰ $\delta^{18}O$ and < -180 ‰ $\delta D$)





and might be influenced by the pooling of meltwater from snow and ice during ice jamming events. The stable isotopes shown in Fig. 5 have been measured in seven sets (separated by dashed vertical lines) and in two different labs (see Table 1). There are no apparent offsets between sample sets. The d-excess of the sample set that was measured at the MPI (#488 to #612) shows less noise compared to previous sets.

**3.3 Dissolved organic carbon concentrations and absorption of colored dissolved organic matter**

For dissolved organic carbon (DOC), the sample water was filtered right after sampling through a 0.45 μm cellulose acetate filter, which had been pre-rinsed with 20 mL of sample. DOC samples were filled into a pre-rinsed 20 mL glass vial and acidified with 25 μL HCl Suprapur (10 M) and stored in the dark at 4°C. After transport, DOC samples #001 to #487 were analyzed at the hydrochemistry laboratory at AWI Potsdam. DOC concentrations were analyzed using high temperature catalytic oxidation (TOC-V$_{CPH}$, Shimadzu). Three replicate measurements of each sample were averaged. After every ten

samples, a blank (Milli-Q water) and a standard was measured. Eight different commercially available certified standards covered a range between 0.49 mg/L (DWNSVW-15) and 100 mg/L (Std. US-QC). The results of standards provided an accuracy no worse than ± 5 %. DOC concentrations for samples #488 to #612 (26 August 2021 to 16 August 2022) were analyzed at Lomonosov Moscow State University (MSU) using a TOPAZ NC, Informanalitika LLC (Russia). Three replicate measurements of each sample were averaged and three standards (5, 15, and 100 mg L$^{-1}$) as well as blanks (Milli-Q water)

were used to ensure high accuracy of the measurements. For the period 10 September 2021 to 31 July 2022, total carbon and inorganic carbon concentrations were measured on unfiltered samples (Fig. C2). For the absorption of colored dissolved organic matter (a$_{CDOM}$(λ)), the samples were filtered right after sampling through a 0.45 μm cellulose acetate filter, which had been rinsed with 20 mL sample water. a$_{CDOM}$(λ) samples were collected into pre-rinsed 50 mL amber glass bottles that were stored in the dark at 4°C until analysis. After transport, a$_{CDOM}$(λ) for samples #001 to #039 (20 April 2018 to 13 September

2018) and #079 to #201 (11 April 2019 to 11 September 2019) were measured at the Otto Schmidt Laboratory for Polar and Marine Research (OSL) in Saint Petersburg, Russia, using a double beam SPECORD 200 (Analytik Jena) spectrophotometer. Samples #043 to #078 (29 September 2018 to 06 April 2019) and #202 to #478 (13 September 2019 to 23 August 2021) were measured at the German Research Center for Geosciences (GFZ) in Potsdam, Germany using a double beam LAMBDA 950 UV/Vis (PerkinElmer) spectrophotometer. Samples #488 to #612 (26 August 2021 to 16 August 2022) were measured at MSU,

Russia using a PE-5400-UV (ECROS LLC) spectrophotometer. The absorbance (A) was measured between 200 and 800 nm in 1 nm steps using a 1, 2, or 5 cm cuvette, depending on the expected concentration of dissolved organic matter. Napierian absorption (a) was calculated from the resulting absorbance measurements via

$$aCDOM(\lambda) = \frac{2.303 * A(\lambda)}{l}, \qquad (2)$$

where l is the path length (length of cuvette in meter). Every five to ten samples, the reference sample (Milli-Q water) was

exchanged and a blank was measured to avoid artifacts from instrument drift.





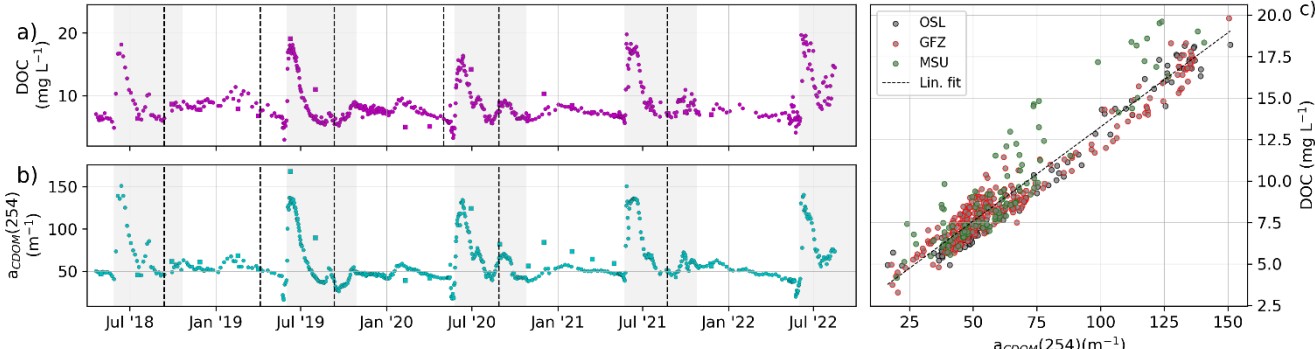

**Figure 6: Time series of (a) DOC and (b) a$_{CDOM}$(254). Dashed black lines separate sample sets and indicate a possible change in measurement protocol and method (see Table 1). (c) Relationship between a$_{CDOM}$(254) and DOC where the dashed black line shows the linear fit of all data. The outline colour of the circles indicates the lab where samples were measured.**

The DOC concentration ranged between 3.1 and 19.8 mg L$^{-1}$ (Fig. 6a). a$_{CDOM}$(254) ranged between 16.4 and 150.9 m$^{-1}$ (Fig. 6b). Low DOC concentrations and a$_{CDOM}$(254) occurred either in the late winter, or in summer during periods of low discharge. The highest annual DOC and a$_{CDOM}$(254) occurred during the spring freshet, when discharge was the highest. a$_{CDOM}$(254) and DOC showed a very strong linear relationship (r²=0.92). The samples measured at MSU showed a significantly lower r² (=0.85), compared to the samples measured at GFZ (r²=95) or OSL (r²=0.98). Note that while we only show a$_{CDOM}$ at the

wavelength 254 nm, the dataset in Juhls et al. (2020b) contains all wavelengths between 200 and 800 nm. Based on DOC and a$_{CDOM}$(λ), we calculated three optical indices as indicators for the chemical composition and molecular structure of the organic matter: Specific Ultraviolet Absorbance at 254 nm (SUVA$_{254}$), the spectral slopes of a$_{CDOM}$(λ) between 275 and 295 nm (S275-295), and the slope ratio (SR). These three indices are reported to correlate with the aromaticity and molecular weight of bulk DOC (Helms et al., 2008; Weishaar et al., 2003). The change in S275-295 has been reported to be a good indicator for

photodegradation of DOM (Fichot et al., 2013; Fichot and Benner, 2012; Helms et al., 2008). SUVA$_{254}$ (m² g C$^{-1}$) was calculated by dividing the decadal absorption A/l (m$^{-1}$) at 254 nm by DOC concentration (mg L$^{-1}$). The SUVA$_{254}$ ranged between 1.41 and 3.89 mg L$^{-1}$ with highest values after the freshet and lowest values during winter and low discharge periods in summer. S275–295 was determined by fitting the data for the wavelength ranges 275–295 nm to the exponential function (Helms et al., 2008):

$$aCDOM(\lambda) = aCDOM(\lambda 0) * e^{-S(\lambda - \lambda 0)}, \tag{3}$$

where a$_{CDOM}$(λ$_0$) is the absorption coefficient at reference wavelength λ$_0$ and S is the spectral slope of a$_{CDOM}$(λ) for the chosen wavelength range. S275-295 ranged between 0.0130 and 0.0182 nm$^{-1}$. The SR was calculated by dividing the spectral slope from 275 to 295 nm by the spectral slope between 350 and 400 nm. SR ranged between 0.814 and 1.36, not including the last set of samples (from 26 August 2021). This set, measured with the PE-5400-UV spectrophotometer at the MSU, shows noisy

results for longer wavelengths visible in the S350-400 that is used to calculate SR. Consequently, SR data for that sample series is not recommended to be used. The results for S275-295 of this sample series, however, are comparable with previous sample series.





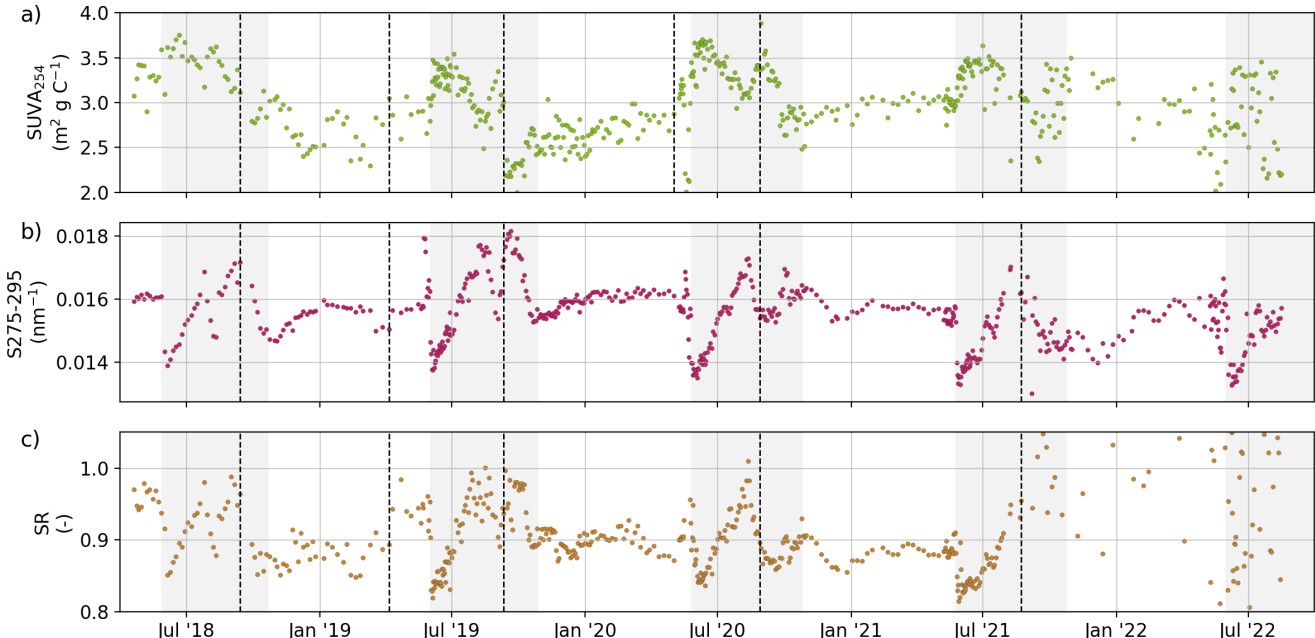

**Figure 7: (a) Time series of SUVA₂₅₄, (b) S275-295 and (c) SR. Dashed black lines separate sample sets and indicate a possible change in measurement protocol and method (see Table 1)**

### 3.4 Fluorescent dissolved organic matter

Fluorescence measurements were carried out at the Technical University of Denmark on a Horiba Aqualog with a 1 cm quartz cuvette (Suprasil grade, Helma GmbH) using the same sample as for $a_{CDOM}(\lambda)$. Fluorescence emission was recorded between 220 and 620 nm (increment ~3.3 nm) at excitation wavelengths between 240 nm and 600 nm (increment 3 nm). The accuracy of the optical components and the immaculacy of cuvettes were validated daily (Wünsch et al., 2015). Fluorescence excitation-emission matrix (EEM) data were processed in Matlab (MathWorks Inc.) using the drEEM toolbox (v0.6.3, Murphy et al. (2013). Inner filter effects were compensated using the absorbance-based method (Kothawala et al., 2013) and the fluorescence counts were converted into Raman units (R.U.) using the water blank's (sealed reference cuvette) Raman emission band at 350 nm. Raman and Rayleigh scatter were removed from each EEM without interpolation.




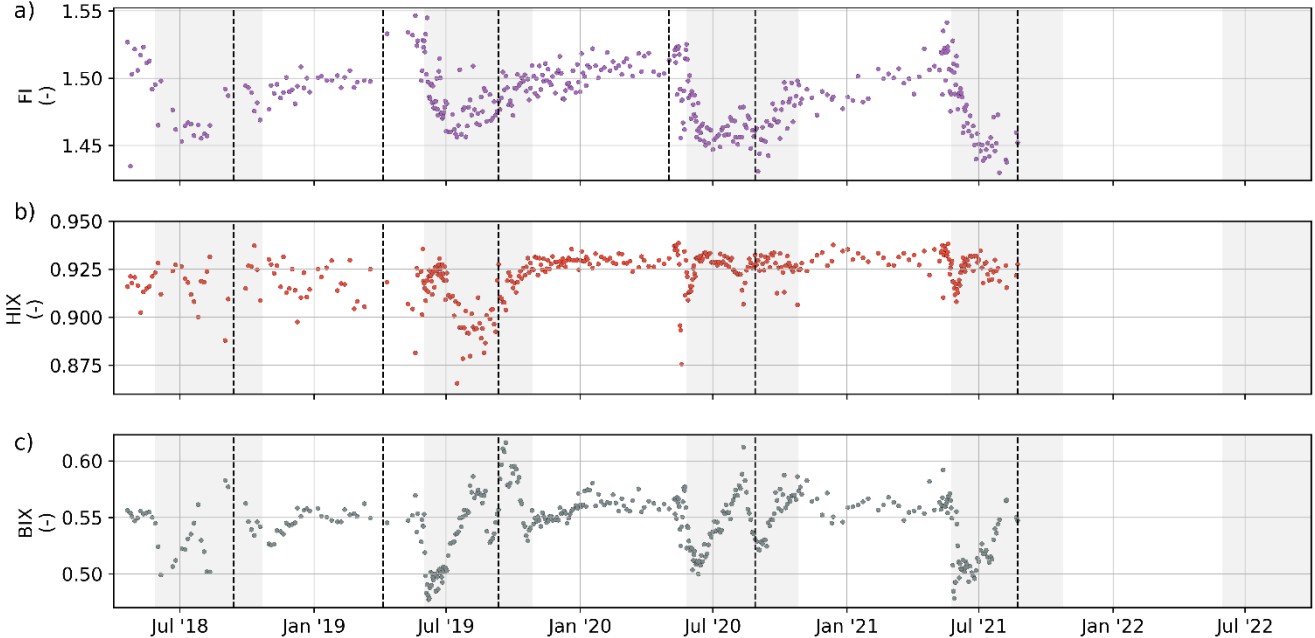


**Figure 8: (a) Time series of Fluorescence Index (FI), (b) Humification Index (HIX) and (c) Biological Index (BIX). Dashed black lines separate sample sets and indicate a possible change in measurement protocol and method (see Table 1).**

Fluorescence-derived DOM optical indices are proxies for the character of DOM with respect to its degree of humification and biological degradation, therefore giving insights into DOM's source (Fig. 8). The Fluorescence Index (FI) was determined as

the ratio between the emission intensities at 470/520 nm for an excitation wavelength of 370 nm (Maie et al., 2006). FI values generally ranged between 1.2 and 2, indicating terrestrial or microbial origin of DOM, respectively (D'Andrilli et al., 2022; McKnight et al., 2001). The FI values for our samples ranged between 1.43 and 1.54 (median 1.48), with the highest values observed just before the ice-free period, decreasing drastically after the ice breaks up. The Humification Index (HIX) estimates the degree of humification of DOM (Zsolnay, 2003; Zsolnay et al., 1999). We calculated it as modified by Ohno (2002) as the

ratio of the areas of two spectral wavelength regions in the emission spectra for an excitation at 254 nm and obtained it as:

$$HIX = \frac{H}{(H+L)}, \tag{4}$$

where H is the area between 435 and 480 nm in the emission spectra and L is the area in the emission spectra between 300 and 345 nm. An increase in the degree of aromaticity (humification) will be associated with higher HIX values. HIX values ranged from 0.78 to 0.93 (median 0.92) showing relatively steady values during the ice-covered season with values decreasing at the

beginning of the ice-free season. The biological index (BIX) is a proxy used to assess the biological modification of DOM. The BIX is obtained by calculating the ratio of the emission at 380 and 430 nm, excited at 310 nm (Huguet et al., 2009):

$$BIX = \frac{IEm380}{IEm430}. \tag{5}$$





High BIX values correspond to autochthonous origin of DOM, i.e., freshly released DOM, whereas low BIX values indicate allochthonous DOM (Huguet et al., 2009). BIX values varied between 0.48 and 0.61 (median 0.54) during the sampling period.

Values presented low variability during the ice-covered periods, followed by a rapid decrease at the beginning of the ice-free season, then rapidly increasing and reaching the highest values. The underlying fluorescence phenomena were distinguished using parallel factor analysis (PARAFAC) with the N-way toolbox algorithms (Andersson and Bro, 2000). Prior to modeling, excitation scans shorter than 250 and longer than 450 nm and emission scans shorter than 312 and longer than 600 nm were deleted to save computation time. EEMs were normalized by division with the 1.2th root of their standard deviation to give

samples with different overall fluorescence a similar leverage. Models with two to eight components were explored. All models were constrained to fit components with positive scores and loadings (i.e. nonnegativity). Models were initialized with random numbers and the best model (with the lowest model error) out of 50 solutions was selected. A maximum of 2500 iterations was allowed and relative change in fit of $10^{-6}$ was chosen as the convergence criterion. Ultimately, the seven-component model was chosen as the most appropriate approximation and its loadings were validated using the split-half approach. Seven

fluorescent DOM components (Fmax) were isolated with PARAFAC (Fig. 9a). Overall, all components presented the same temporal patterns with relatively steady values during the ice-covered season, followed by a decrease just before the ice break-up and rapidly increasing and quickly reaching the highest values shortly after ice break-up (Fig. 9b). Components 1 and 2 (C1 and C2, respectively) showed fluorescence peaks in the UV range, which are generally associated with autochthonous DOM such as protein-like compounds (Coble, 2007). The fluorescence intensities of C1 and C2 varied from 0.04 to 0.54 R.U. and

from 0.03 to 0.3 R.U., respectively. C3 to C7 showed a fluorescence peak in the visible wavelength range (>400 nm), which is generally associated with terrestrial humic-like compounds (Coble, 2007). C3 and C4 showed the highest fluorescence intensities, varying from 0.18 to 1.38 R.U. and from to 0.08 R.U. to 2.03 R.U., respectively. C5 and C6, ranged from 0.30 to 2.8 R.U. and from 0.06 to 0.62 R.U., respectively, whereas C7 values ranged between 0.12 and 1.05 R.U..

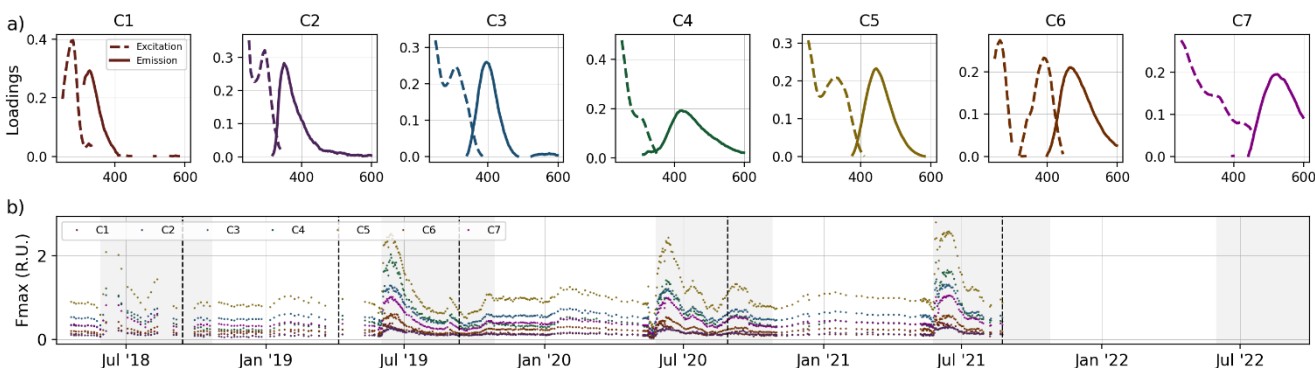

**Figure 9: (a) Loading excitation and emission spectra of the seven identified PARAFAC components. (b) Time series of the Fmax loadings for the seven components. Dashed black lines separate sample sets.**



### 3.5 DOC radiocarbon

Samples for DOC $\Delta^{14}$C analysis were taken biweekly from 30 September 2019 to 15 July 2021. For each sample, an acid-washed 250 mL HDPE bottle was rinsed two times with river water before filling it with the sample in order to preempt contamination from sources such as outboard motor exhaust or dust particles. Water was collected upstream from the boat and operator to ensure the utmost integrity of the samples. After sample collection, each bottle was promptly sealed and kept frozen at a temperature of -20°C during subsequent transportation and storage. Radiocarbon contents of DOC were analyzed using a MIniature CArbon DAting System (MICADAS) at Alfred Wegener Institute in Bremerhaven, Germany and methods described in Mollenhauer et al. (2021). In the laboratory, water samples were thawed and filtered over 0.75 µm glass fiber filters (GF/F, Whatman). Dissolved organic matter (DOM) was concentrated by evaporation of water using a rotary evaporator. Concentrated DOM was transferred into silver liquid cups (70 µL, Elementar part#: 200010387) and dried completely on a hot plate followed by storage in a dessicator kept at 40°C. After drying, silver cups were folded into small packages and combusted in an elemental analyzer (EA, Elementar varioIsotope), coupled to the Ionplus Gas Interface System (GIS; Wacker et al. (2013), allowing the transfer of $CO_2$ directly into the hybrid ion source of the MICADAS. Samples were analyzed as gas for a 12-minute measurement cycle, data were evaluated using the BATS software package (Wacker et al., 2013) and normalized against Oxalic Acid II standard gas ($CO_2$ produced from Oxalic Acid II, NIST SRM4990C) and blank corrected against $^{14}$C-free $CO_2$. Secondary blank correction was performed using process blank determination according to the method of Sun et al. (2020), and errors were propagated following Wacker and Christl (2011). Results are reported as $^{14}$C, conventional radiocarbon ages (years BP) (Stuiver and Polach, 1977).

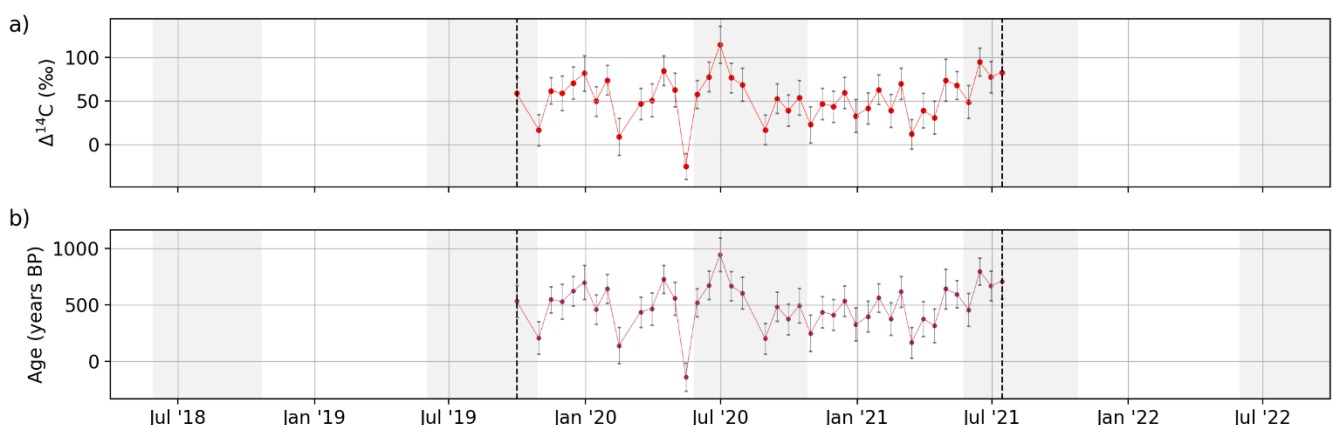

**Figure 10:** (a) $\Delta^{14}$C of DOC and the 2 x standard deviation (2σ) shown as error bars, (b) corresponding age and the 2 x standard deviation (2σ) shown as error bars. Dashed black lines indicate the start and end date of the set of samples that were analyzed.

No clear seasonal patterns were identified in the radiocarbon contents of DOC, with $\Delta^{14}$C values ranging between -25 ‰ and 115 ‰ (Fig. 10a) and corresponding conventional radiocarbon ages between -945 and 138 years (Fig. 10b). The lack of clear seasonal differences may be due to the lower number of samples and the reduced sampling period.





### 3.6 Nutrients

Samples for dissolved inorganic nutrient (ammonium ($NH_4$), nitrite ($NO_2$), nitrate ($NO_3$), phosphate ($PO_4$), and silicate (Si)) analysis as well as for total dissolved nitrogen and total dissolved phosphorus (TDN, TDP) were filtered through a 0.45 µm cellulose acetate filter, which had been rinsed with 20 mL sample water. Nutrients/TDN/TDP samples were filled into pre-

rinsed 20 mL PE bottles and stored frozen at -18°C until measured. For TN/TP (total nitrogen and total phosphorus), unfiltered samples were filled into pre-rinsed 20 mL PE bottles and stored frozen at -20 °C until measured. TN/TDN and TP/TDP for samples #001 to #039 (20 April 2018 to 13 September 2018) and dissolved inorganic nutrients for samples #001 to #201 (20 April 2018 to 11 September 2019) were measured at the Otto Schmidt Laboratory in St. Petersburg, Russia. Dissolved inorganic nutrients were analyzed on an automated continuous flow system (San++, SKALAR, Netherlands) with standard

colorimetric techniques (Aminot et al., 2009). For the determination of TN/TDN and TP/TDP, the persulfate oxidation method (Knapp et al., 2005) was used. The first step was the oxidation of total dissolved nitrogen (TDN, the sum of nitrate, nitrite, ammonium and dissolved organic nitrogen (DON)) to nitrate and TDP to phosphate. Therefore, 24 ml of the sample plus 2 ml of persulfate oxidizing reagent (POR) was added to a Teflon bottle. The POR contained ACS-grade sodium hydroxide, certified ACS-grade boric acid and certified ACS-grade potassium persulfate, which was recrystallized three times (Hansen and

Koroleff, 2007). The digestion was performed by autoclaving at 121°C. In the same digestion also the total phosphorus was measured. Reagent blank was below 0.1 µM. Environmental matrix reference materials (Environment and Climate Change Canada) were used as tracking standards in every batch of samples. TN and TP for samples from #201 to #362 and dissolved inorganic nutrients for samples #202 to #487 were measured at the Helmholtz-Zentrum Hereon in Geesthacht, Germany. Nutrient concentrations were analyzed in duplicates using an automated continuous flow system (AA3, Seal Analytical,

Germany) and standard colorimetric techniques (Hansen and Koroleff, 2007). Detection limits were 1 µM for nitrate and silicate, 0.5 µM for nitrite, ammonium and phosphate. For the determination of TDN/TDP, we used the same method as described above for the previous samples. For digestion, a microwave (CEM, Mars 5) was used. The reagent blank was always <2 µM. As reference, internal standards of ammonium sulfate and urea were used. In the same digestion, total phosphorus was measured. Reagent blank was below 0.1 µM. For the analysis of TDN and TNP, we used the same method as was used for

unfiltered water samples (total). The results of standards provided an accuracy better than ± 10 %. TN/TDN, TP/TDP for samples #488 to #612 were measured at the MSU using a PE-5400UV spectrophotometer by ECROS LLC (Russia). Frozen samples were thawed at room temperature in bulks of 10 to 20 samples at once and processed the same day as soon as they reached the appropriate temperature. Concentrations of inorganic phosphorus (orthophosphate) in filtered and unfiltered samples were determined photometrically by the Murphy-Riley method (molybdenum blue reaction of orthophosphate with

ammonium molybdate and antimony potassium tartrate in sulphuric acid, reduced by ascorbic acid, measured at wavelength 885 nm in a 5 cm cuvette). Approved detection range of the specific method used was 0.005-0.100 mg/L of orthophosphate (or 0.0016-0.033 mg P/L), standard error at p=0.05 was 0.0001+0.08*X, where X is the determined concentration in mg P/L. To account for turbidity, two measurements of optical density were made: 1) without ascorbic acid, 2) with ascorbic acid. Total




phosphorus concentrations were determined after persulphate digestion (samples heated in an autoclave at 121°C for 1 hour with ammonium persulphate added) by the same method as for inorganic phosphorus. TN concentrations were determined by alkaline persulphate digestion (samples autoclaved at 120°C for 90 minutes with potassium persulphate and sodium hydroxide, then measured in a 1 cm quartz cuvette at 207 nm wavelength after adding sulphuric acid). Approved detection range of the specific method used was 0.1-6.0 mg N/L, standard error at p=0.05 is 0.04+0.077*X, where X is the determined concentration in mg N/L. Calibration coefficients for the spectrophotometer were obtained by running the described procedures on solutions

diluted from a standard 0.5 g/L solution of phosphate ion and 0.5 g/L solution of TN.

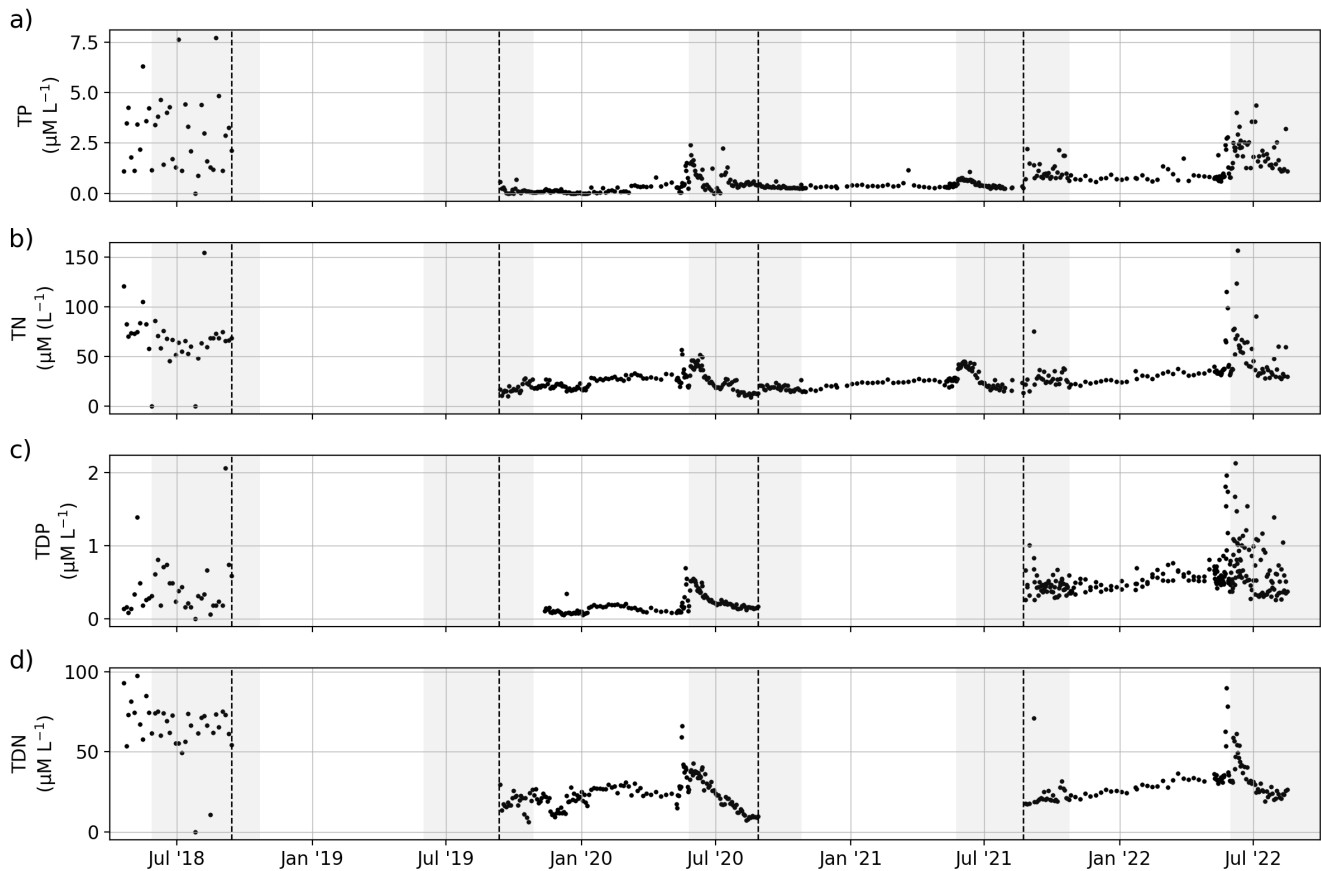

**Figure 11: Concentrations of (a) total phosphorus, (b) total nitrogen, (c) total dissolved phosphorus, and (d) total dissolved nitrogen. Dashed black lines separate sample sets and indicate a possible change in measurement protocol and method (see Table 1).**

TN/TDN, TP/TDP showed peaks during the freshet in June and decreasing concentrations throughout the summer. The

difference in noisiness of the data between the four available sample sets indicate varying quality of the analysis.





**Figure 12: Concentrations of dissolved inorganic nutrients: (a) Si, (b) PO₄, (c) NH₄, (d) NO₂, (e) NO₃, (f) NO₂+NO₃ determined by colorimetric/photometric methods (black) and by ion chromatography (red). Dashed black lines separate sample sets and indicate a possible change in measurement protocol and method (see Table 1).**

PO₄, NH₄ and NO₂ showed annual peak concentrations during the freshet in June whereas NO₃ showed a gradual increase during the winter and a sudden decrease during the freshet. Si concentrations increased from the freshet until mid/end-winter.



### 3.7 Dissolved elemental and ion concentrations

Samples #079 to #487 (6 April 2019 to 23 August 2021) analyzed for concentrations of major dissolved elements and ions were frozen untreated directly after sampling. After transport, samples were thawed (24h at room temperature) in the

hydrochemistry laboratory at the hydrochemistry laboratory at AWI in Potsdam, Germany. They were then filtered using a syringe-mounted 0.45 μm cellulose acetate filter and kept cool and dark until analysis. Samples for the first year (20 April 2018 to 6 April 2019; sample #001 to #078) were filtered right after sampling and transported cool and dark. Samples #079 to #487 were frozen right after sampling and thawed and filtered after transport prior to analysis. Concentrations of major ions (Sulfate ($SO_4$), bromide (Br), nitrate ($NO_3$), phosphate ($PO_4$), chloride (Cl), and fluoride (F)) were determined using ion

chromatography (Thermo-Fischer ICS 2100; Weiss, (2001)). A blank (Milli-Q water) and a standard were measured every ten samples. A commercially available certified standard in two different dilutions (1:5 and 1:10) was used to determine measurement accuracy and the detection limits (Table 1). Total dissolved elemental concentration (aluminum (Al), barium (Ba), calcium (Ca), iron (Fe), potassium (K), magnesium (Mg), manganese (Mn), sodium (Na), phosphorus (P), silicon (Si), and strontium (Sr)) samples #001 to #078 were filtered right after sampling and cooled. Samples #079 to #487 were frozen

right after sampling and thawed after transport and filtered prior to analysis. Then, samples were acidified with 65 % HNO3 (65 % suprapur) and were measured with inductively coupled plasma optical emission spectroscopy (ICP-OES; Perkin Elmer Optima 8300DV; Boss and Fredeen, (1997)). For the samples #202 to #287 (13 September 2019 to 2 May 2020), we compared elemental and ion concentrations from two sets of samples that were 1) frozen right after sampling and thawed and filtered only after transport, and 2) filtered right after sampling and transported unfrozen (cooled at +4°C). Comparing the results of

the two sets shows that differences in sample processing affect elemental and ion concentrations in ways that introduce systematic biases (Fig. B2 & B3). These biases differ in magnitude and direction depending on parameter, but for most parameters freezing results in lower concentrations. Samples #488 to #612 (26 August 2019 to 16 August 2021) were treated identically to the samples #079 to #487, but measured at the MSU. Concentrations of major dissolved elements and ions were measured using ion chromatography using a Concise ICSep An2 column for major ions (Cl, $SO_4$, F, $NO_3$) and a Shodex IC

YS-50 column for total dissolved elemental concentration (Na, K, Mg, Ca, Si, $NH_4$). Most dissolved elemental and ion concentrations ($SO_4$, $NO_3$, Ba, Cl, F, Ca, K, Mg, Na, Si, Sr) increased during the winter (similar to EC, Fig. 4) and decreased sharply during the freshet (Fig. 13 & 14). Fe and Mn showed a strong peak and their annual maximum during the freshet. Al and P did not indicate clear seasonal patterns.



**440** **Figure 13: Concentrations of major ions: (a) Si, (b) Br, (c) NO₃, (d) PO₄, (e) Cl, and (f) F. The horizontal red line shows the detection limit. Dashed black lines separate sample sets and indicate a possible change in measurement protocol and method (see Table 1).**

In addition, for the samples #001 to #078, the germanium (Ge) concentration and Si isotope composition ($\delta^{30}$Si) was measured (Fig. C1). Heavy metals (Pb, Cr, V, Co, Ni, Cu, Zn) were measured for samples #202 to #487. Concentrations for these





parameters and for all samples were below detection limit, with the exception of some zinc concentrations between 20 and 196

µg $L^{-1}$.







**Figure 14: Concentrations of total dissolved elemental concentrations: (a) Al, (b) Ba, (c) Ca, (d) Fe, (e) K, (f) Mg, (g) Mn, (h) Na, (i) P, (j) Si, and (k) Sr. The horizontal red line shows the detection limit. Dashed black lines separate sample sets and indicate a possible change in measurement protocol and method (see Table 1).**

## 4 Data availability

The raw data and all related metadata are stored at the Alfred Wegener Institute (AWI), Germany. Final aligned and cleaned datasets are available on Pangaea: https://doi.org/10.1594/PANGAEA.913197 (Juhls et al., 2020b). Detailed metadata are available with digital object identifiers (DOI), including the principal investigator's contact information. For specific questions, please contact the principal investigator associated with the parameter. In addition, the data from this collection can be explored in an interactive dashboard at https://lena-monitoring.awi.de/.

## 5 Conclusion

The dataset presented here is the result of a comprehensive, year-round, high-frequency monitoring of the biogeochemistry of the Lena River, covering nearly 4.5 years. The data collection includes a wide range of biogeochemical parameters, maintaining consistency for most parameters in coverage and data quality. This consistency is achieved through the involvement and committed engagement of local partners, simple sampling and sample handling protocols, and effective real-time communication between sampling personnel and scientists. To the best of our knowledge, this data collection represents the most extensive and detailed coverage of an Arctic river's biogeochemistry to date. The high-frequency nature of this dataset is particularly significant, allowing for the observation of biogeochemical changes occurring on a weekly or shorter time scale and justifying interpolation between sampling dates. The high-frequency sampling eliminates the need for gap-filling, e.g. using load models, which ignore flux processes that are independent of discharge. Studies, based on this dataset show improved estimates of the loads supplied from terrestrial sources into the Arctic Ocean in terms of both magnitude and timing (Juhls et al., 2020a; Sanders et al., 2022). This dataset also improves the understanding of freezing processes (Lütjen et al., 2024) and the role of the ice-covered period for biogeochemical processing of nutrients in the Lena River prior to their delivery to the Arctic Ocean (Opfergelt et al., in review). Furthermore, these data serve as a valuable resource for satellite data validation, as demonstrated by El Kassar et al. (2023). Other potential applications of this dataset include enhancing climate and earth system models and supporting policy decisions regarding Arctic environments. This dataset establishes a baseline to monitor future environmental changes across various time scales, from precipitation events to seasonal and interannual variations as well as to detect future changes in river hydrology. Given the current geopolitical situation limiting international access to the Russian Arctic, such robust, long-term datasets will be of great importance to monitor and understand ongoing environmental changes. Overall, the breadth and quality of this dataset provide an invaluable foundation for future research and monitoring efforts in the rapidly changing Arctic region.





**Appendix A**

The temperature and electrical conductivity of the entire Lena River water column was measured in different years and seasons (August 2016, March to April and August 2019) at a number of locations in different channels within the Lena River Delta.
These profiles show a well-mixed and unstratified water column (**Fig. A1**).

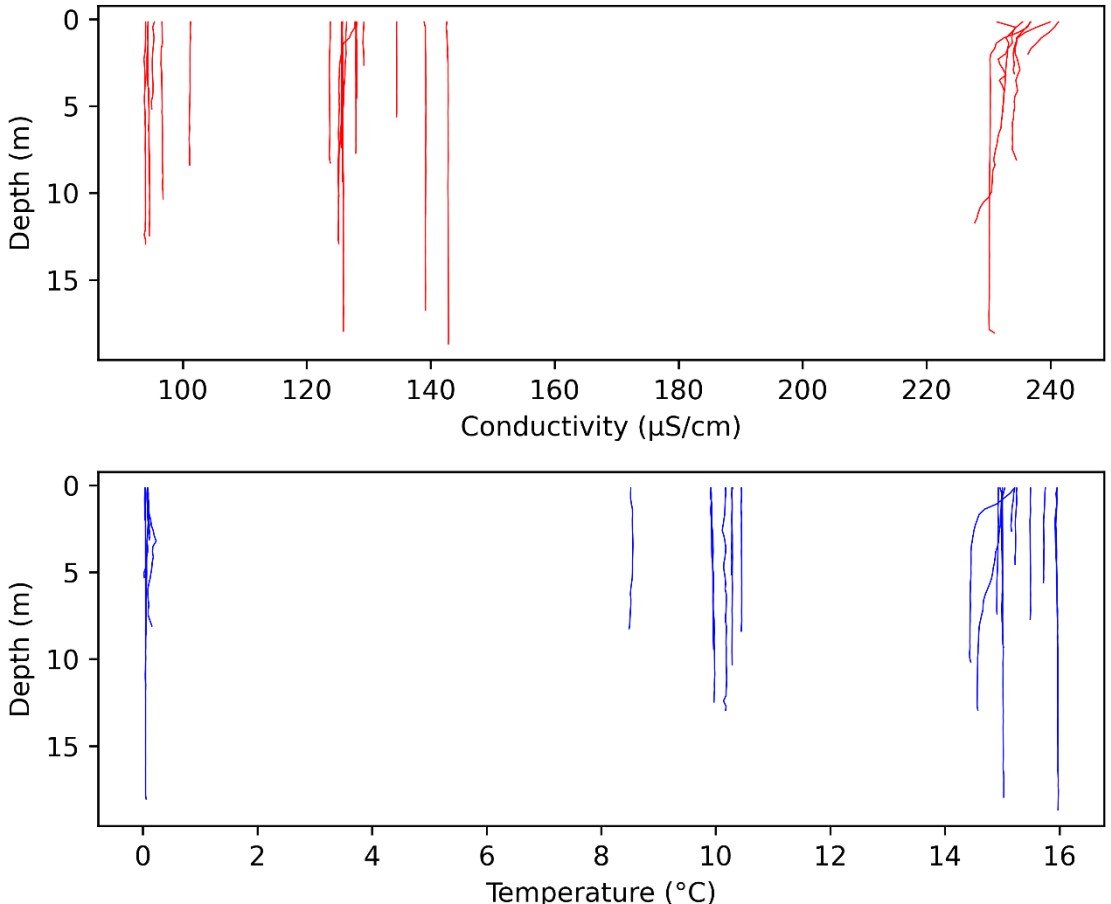

**Figure A1: CTD profiles within channels of the Lena River across different years and seasons (https://doi.pangaea.de/10.1594/PANGAEA.933182 and Overduin et al. (2017)).**

**Appendix B**

For a set of samples covering the period from 13 September 2019 to 2 May 2020, we measured the electrical conductivity (Fig. B1), major ion concentrations (Fig. B2), and dissolved elemental concentrations (Fig B3) with two different protocols to assess the impact of sample processing on the dissolved elemental and ion concentrations. While some dissolved elemental and ion concentrations show minor differences when comparing the results of the different protocols, some others show large

differences or even seasonal differences. For many dissolved elemental and ion concentrations, frozen samples show lower

concentrations compared to unfrozen.

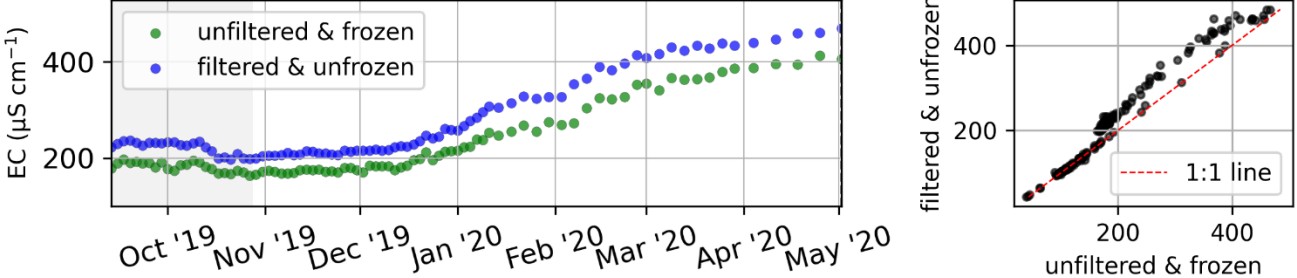

**Figure B1: Comparison of electrical conductivity of two sample sets that were processed with two different protocols. One set was frozen right after sampling, then thawed and filtered after storage and transport (green), and the other set was filtered right after sampling and stored and transported unfrozen/cooled (blue). The analysis method for both sets were identical (see Table 1).**




**Figure B2: Comparison of major ion concentrations of two sample sets that were processed with two different protocols. One set was frozen right after sampling, then thawed and filtered after storage and transport (green), and the other set was filtered right after sampling and stored and transported unfrozen/cooled (blue). The analysis method for both sets were identical (see Table 1).**





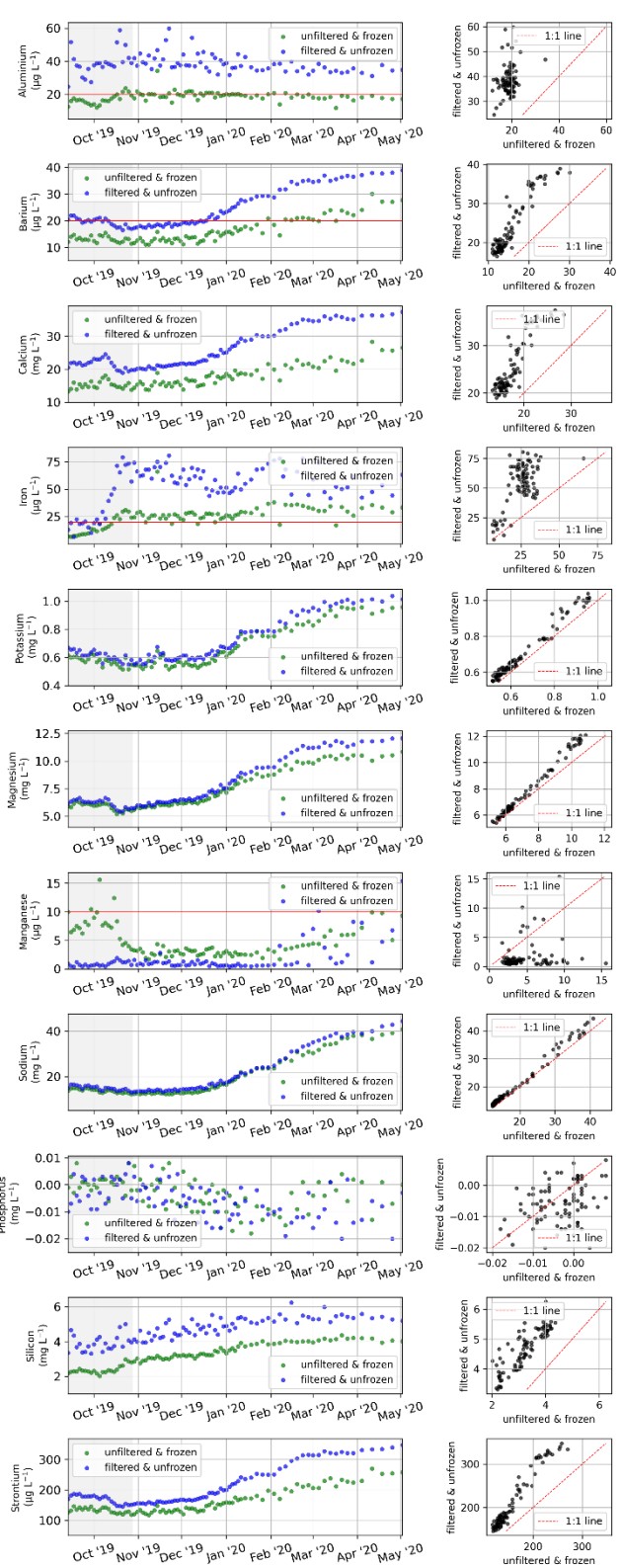





**Figure B3: Comparison of total dissolved elemental concentrations of two sample sets that were processed with two different protocols. One set was frozen right after sampling, then thawed and filtered after storage and transport (green), and the other set was filtered right after sampling and stored and transported unfrozen/cooled (blue). The analysis method for both sets were identical (see Table 1).**

**Appendix C**

In addition to the parameters shown in the main manuscript, some sample sets or rest volumes of samples were used to measure additional parameters, however, not for the entire period of the sampling program. The germanium (Ge) concentration for samples #001 to #077 (20 April 2018 to 28 March 2019) was determined on rest volumes of the samples for total dissolved elemental concentration by ICP-mass spectrometry (ICP-MS, ICAPQ Thermo Fisher Scientific, Earth & Life Institute, UCLouvain, Belgium). The detection limit for Ge is 0.04 nM and the analytical precision of the measurement is ± 8 % for Ge

concentrations < 0.013 nM and ± 4 % for Ge concentrations > 0.013 nM. The silicon isotope composition ($\delta^{30}$Si) was analyzed by MC-ICP-MS (Neptune Plus™ High Resolution Multicollector ICP-MS, Thermo Fisher Scientific, Earth & Life Institute, UCLouvain, Belgium) in wet plasma mode after Si separation using a two-stages column chemistry procedure using an anion exchange resin (Biorad AG MP-1) followed by a cation exchange resin (Biorad AG50W-X12). The instrumental mass bias was corrected using the standard-sample bracketing technique and an external Mg doping. The $\delta^{30}$Si compositions are

expressed in relative deviations of $^{30}$Si/$^{28}$Si ratio from the NBS-28 reference standard using the δ-notation (‰) as follows: $\delta^{30}Si = [(^{30}Si/^{28}Si)sample/(^{30}Si/^{28}Si)NBS-28 -1] \times 1000$. Each single δ-value represents one sample run and two bracketing standards. The $\delta^{30}$Si values are reported as the mean of isotopic analyses from multiple analytical sessions at least in duplicate. The long-term precision and accuracy is ± 0.08 ‰ (SD).

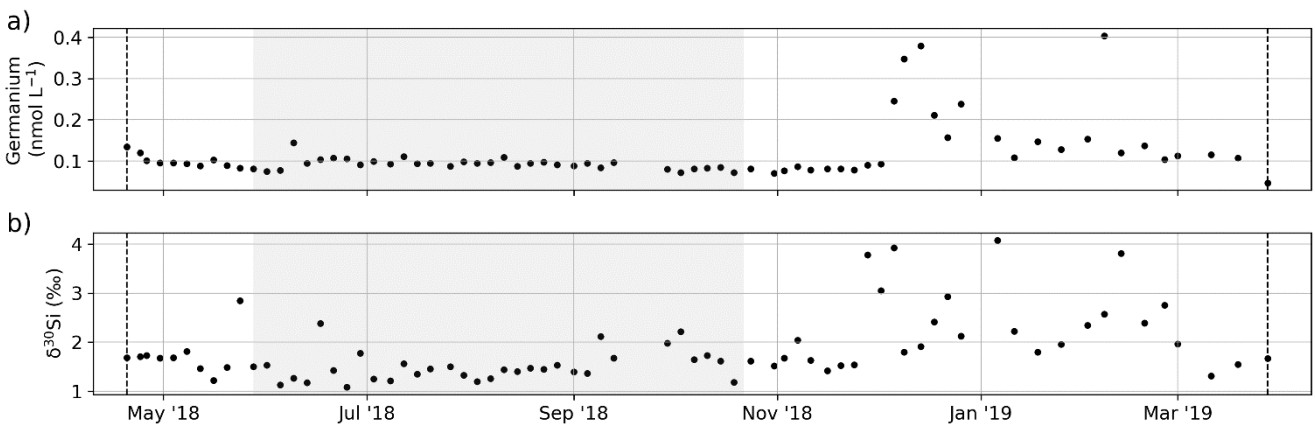

**Figure C1: (a) Germanium concentration, (b) dissolved Si isotope composition. Dashed black lines indicate the start and end date of the set of samples that were analyzed.**

Organic and inorganic carbon concentrations were measured on unfiltered samples for the period 10 September 2021 to 31 July 2022 (Fig. C2) at the Lomonosov Moscow State University in Moscow, Russia (MSU), using a TOPAZ NC manufactured by Informanalitika LLC (Russia). Analyses for the determination of total carbon concentration are based on ISO 8245 for the



determination of the sum of organically and inorganically bound carbon, including elemental carbon, based on infrared spectrometry of $CO_2$ after thermocatalytic oxidation of all carbon species. Analyses for the determination of inorganic carbon concentration are based on ISO 8245 for the determination of sum of elemental carbon, carbonates and bicarbonates, carbon dioxide and monoxide, cyanide, cyanate and thiocyanate based on infrared spectrometry of $CO_2$ after oxidation with phosphoric acid.

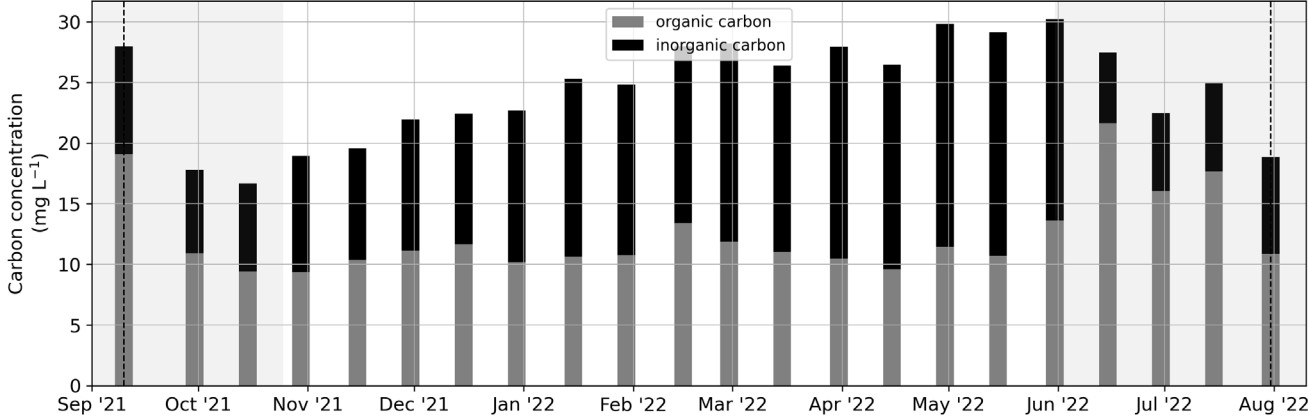


**Figure C2: Organic (grey bars) and inorganic carbon (black bars) concentration measured on unfiltered samples that were frozen right after sampling. The sum of organic and inorganic shows the total carbon concentration (top of the stacked bars).**

**Appendix D**

Comparisons to data from the The Arctic Great Rivers Observatory for which the water sampling is conducted in Zhigansk
(Fig. 1a), approximately 700 km upstream of the Research Station Samoylov Island.

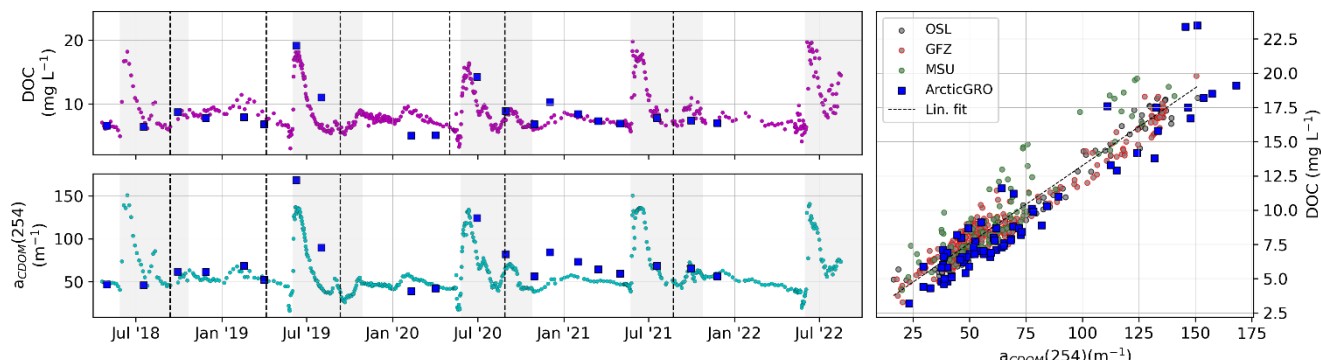

**Figure D1: Comparison of DOC concentration and absorption by CDOM at 254 nm with data collected by PARTNERS and ArcticGRO.**



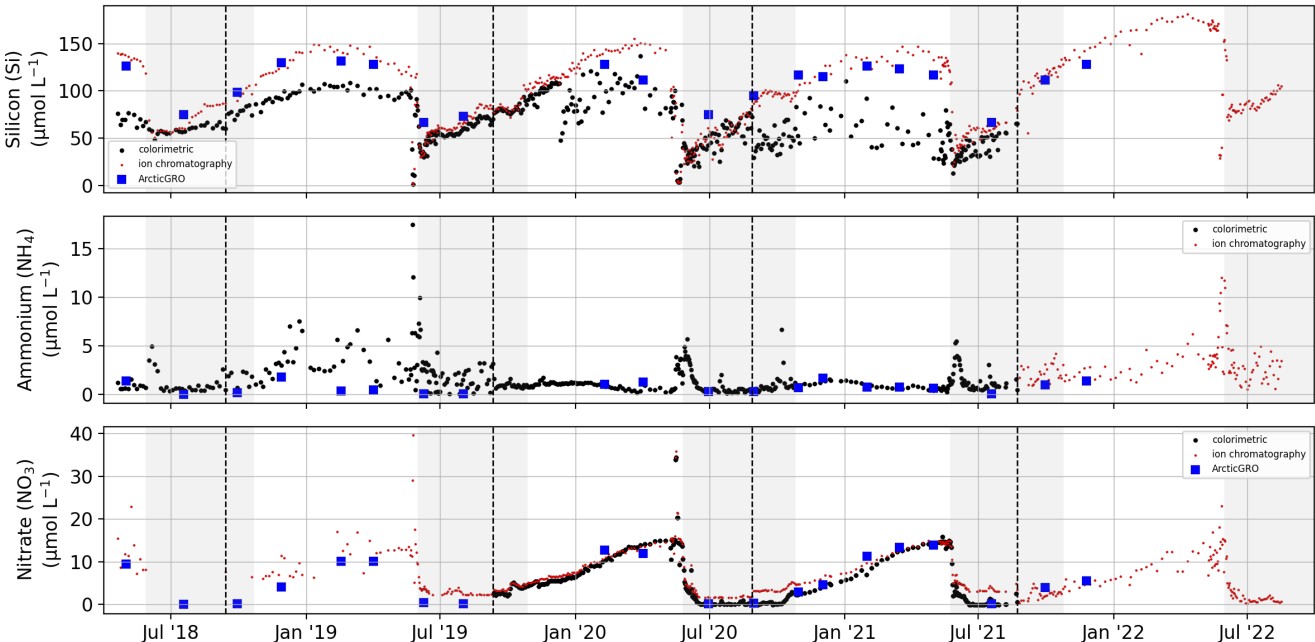

**Figure D2: Comparison of Si, NH₄, and NO₃ concentration with data collected by PARTNERS and ArcticGRO.**

**Author contributions**

The research was conceived and coordinated by BJ, AM, and PPO, with initial support from JH and BH. AE was responsible for the preparation of consumables and laboratory analysis of DOC and ions at AWI Potsdam. SA managed program communication and logistics between Russia and Germany. EE, FG, and FM created the data dashboard, processed the data,

and ensured quality control. SC, MT, and OE conducted DOC, CDOM, ions, and nutrients measurements at MSU Moscow. Stable oxygen and hydrogen isotope analyses were performed by HM at AWI Potsdam and NT at MPI Yakutsk. Sample and consumables transport in Russia were overseen by GTM, LL, EF, and VP. Sampling was supported by EA, LL, and AC. Nutrient measurements were carried out by TS at Hereon. DOC radiocarbon measurements were performed by HG and GM at AWI Bremerhaven. CDOM and nutrient measurements at OSL, St. Petersburg, were conducted by VP, AC, and JH. Ge and

Si isotope measurements were carried out by SO. BJ piloted the creation of figures and tables. BJ wrote the main content of the paper, with further contributions and suggestions from all co-authors.

**Competing interests**

At least one of the (co-)authors is a member of the editorial board of Earth System Science Data.



**Financial support**

BJ was supported by the EU Horizon 2020 program (Nunataryuk, grant no. 773421), the European Space Agency (ESA) as part of the Climate Change Initiative (CCI) fellowship (ESA ESRIN/Contract No. 4000l3376l/2l/I-NB) and by the BNP Paribas Foundation (FLO CHAR Project). Funding for the MICADAS radiocarbon laboratory was provided through AWI institutional core funding and HG was funded by the MARUM Cluster of Excellence "The Oceans Floor – Earth's Uncharted Interface" (DFG Project number 390741603). Funding was provided to SO by the European Research Council (ERC) under the European

Union's Horizon 2020 research and innovation programme (WeThaw n°714617) and by the Fonds National de la Recherche Scientifique (FNRS, FC69480). LL, NT and GM were funded by the Melnikov Permafrost Institute SB RAS (project numbers 122012400106-7 and 122011800064-9). SC has been supported by the Kazan Federal University Strategic Academic Leadership Program ("PRIORITY-2030") and Ministry of Science and Higher Education of Russian Federation under the Agreement No 075-15-2024-614.

**Acknowledgement**

We would like to express our sincere gratitude to the staff of the Research Station Samoylov Island for their central role and expertise in carrying out in situ measurements, water sampling, logistics and communications: Fedor Sellyakhov, Sergey Volkov, Andrey Astapov, and Viktor Zykov. We are grateful to Mikaela Weiner and Andreas Marent at AWI for their work on stable oxygen and hydrogen isotopes. Our appreciation goes to Volkmar Assmann and Waldemar Schneider for their

logistical support in Russia through AWI. Special thanks to Pia Esterl, Martha Lütjen, Henrike Walther, Juan Sebastian Valencia Velasquez, and Caroline Herff at AWI for their contributions to sample processing, analysis, and data processing. We acknowledge the administrative and logistical support provided by Mikhail N. Grigoriev at MPI, Yakutsk, Russia and Luidmila Pestryakova at the North-Eastern Federal University in Yakutsk, Russia. We are thankful to Colin Stedmon, Signe Melbye Andersen, and Anders Dalhoff Bruhn Jensen at DTU for their assistance with FDOM measurements. Finally, we

extend our gratitude to Elizabeth Bonk, Silla M. Thomsen, and Torben Gentz for their laboratory support at AWI MICADAS. We also extend our thanks to Mathias Bochow and Carsten Neumann at GFZ for their support with CDOM measurements.

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
