# Peer review of "Lena River biogeochemistry captured by a 4.5-year high-frequency sampling program"

_Earth System Science Data, 2024_

## Author Comment (AC1)

**We would like to thank the anonymous referees for their thorough review of our manuscript. Please find our responses to each of the referee's suggestions and specific comments below.**

**Font legend:**

Referee comments

Authors response

Changed text in manuscript

**Referee 1**

The work is within the scope of the journal; however, the authors have to invest a bit more to provide it a clear added value. The authors used adequate procedure for sampling, handling and analyses. The manuscript reports highly impressive number of sampling points and various hydrochemical parameters. It clearly presents a valuable data set. However, the need for such a dataset should be clearly explained and justified. As a minimal starting point in the Introduction, please explain the difference (and novelty) of this work compared to data available from Partners/Arctic GRO at this river terminal gauging station.

Thank you very much for your review of our manuscript. Regarding the justification for the need of such a dataset, we have added a sentence into the introduction:

"Understanding the impact of climate shifts requires a high-quality, high-frequency dataset to assess current conditions and predict future trends."

In multiple sentences in the conclusion, we detail ways in which the scientific utility of the dataset is justified. Other than that, we think there is sufficient justification already in the introduction for such a dataset as the one that we present. Here a few examples from our introduction that serve as a justification for the need of this dataset: a) "There is no paleo-historical analogue for these changes, therefore, establishing a baseline of current fluxes and understanding how the system is changing are necessary to anticipate the scope and consequence of future impacts of climate warming and permafrost thaw."; b) "To understand the changes underway, their impacts on the river system and, in turn, their impacts on the global climate, a baseline of observations that includes biogeochemistry is required. It is a prerequisite for deriving improved insights into linkages between land and ocean and between river system and climate that will allow for better constraining future impacts of

continued warming."; c) "Higher sampling frequency can improve annual flux estimates, as does dedicated sampling over the whole hydrological cycle. Arctic rivers are typically characterized by a nival hydrological regime, and, thus, the strong seasonality and high variability in summer water balance may mandate high-frequency data collection, especially during the highly dynamic shoulder seasons (freshet, freeze-up)."; d) "In addition, higher frequency or even continuous in situ measurements (e.g., Castro-Morales et al., 2022) will create new opportunities to validate remotely sensed data (El Kassar et al., 2023) or model results (e.g. Rawlins and Karmalkar, 2024) and to potentially upscale data spatially."

In the main text and figures, the authors should provide a comparison with data of ARCTIC GRO/Partners obtained at the Kysur gauging station. Appendix D is just great, but it should be in the main text. Other measured parameters (those available from ArcticGRO) should be shown as well.

We moved the figures of the Appendix D to the main manuscript (replaced the old ones with the one containing ArcticGRO data) and added a few sentences comparing the data. In addition, we added the remaining parameters that are sampled by ArcticGRO and our program for a direct comparison (temperature, DOC and CDOM, TDN, several ions). Consequently, we removed Appendix D.

We added a sentence to the methods:

"In addition to the data sampled at Samoylov Island, we included data from the ArcticGRO program for all parameters that were measured by both programs (temperature, DOM, nutrients, ions)."

as well as description to the results:

"DOC and aCDOM(254) generally agrees with data from ArcticGRO sampled several hundreds of km further upstream, however it shows a generally lower DOC to aCDOM(254) ratio compared to our data."

"Comparing TDN, Si, $NH_4$, and $NO_3$ with data from ArcticGRO reveals a good agreement between the datasets."

"We compared some of the dissolved elemental and ion concentrations with those measured by the ArcticGRO program, which shows a generally good agreement. Some stronger

differences might be related to the earlier arrival of changing seasons at the ArcticGRO sampling location further south."

We also added the ArcticGRO description to the figure captions.

The authors possess both discharges and concentrations. Export fluxes (via, for example, LOADEST or any other mean) should be calculated and compared with earlier fluxes. It is the duty of the authors to provide the fluxes, the readers cannot do it themselves. Within the concept of this journal, I assume no discussion of concentration dependence on the discharge and comparison with other rivers are needed. However, the export fluxes (mean multi-annual values or yields) should be there.

Thank you for these suggestions to calculate and report fluxes. We do report the selected parameter fluxes from this dataset in research papers some of which are already published. See e.g. DOC and CDOM fluxes in Juhls et al. 2020 (https://www.frontiersin.org/articles/10.3389/fenvs.2020.00053/full), seasonal and interannual DOC fluxes from multiple years compared with satellite-derived fluxes in El Kassar et al., 2023 (https://www.frontiersin.org/journals/marine-science/articles/10.3389/fmars.2023.1082109/full), and nutrient fluxes in Sanders et al., 2021 (https://link.springer.com/article/10.1007/s13280-021-01665-0). We also refer to these papers in the manuscript. While we have additional publications in preparation that will cover annual and seasonal biogeochemical fluxes, we believe this is beyond the scope of ESSD for this manuscript due to methodological considerations in flux calculation. For example, although the load models such as LOADEST are commonly used for estimating fluxes, we find it less suitable for capturing seasonal or long-term fluxes influenced by non-linear processes, such as permafrost thaw, where no consistent relationship between discharge and concentration of biogeochemical parameters exists. To address such complexities, we prioritize high-frequency sampling, which allows for direct linear interpolation between sampling days for daily flux calculations. Currently, we are working on a research paper manuscript that shows differences between load-models trained with a low number of samples per year and a high-frequency sampling program. It is critical that data-users match their flux calculation methods to the purpose for which the fluxes are calculated.

Specific issues:

L90-91 Please note that annual fluxes of most solutes in Arctic rivers can be reasonably (within 20-30 %, which is lower than annual inter-variations) can be approximated by July-August sampling (see for instance https://doi.org/10.1016/j.chemgeo.2022.121180)

Thank you for this note. We do not fully agree with the statement that "most" solutes can be reasonably estimated due to their robust relationship with discharge. The study by Pokrovsky et al., 2022 also shows that there are multiple groups of solutes and biogeochemical parameters that have differently robust or have no relationship to discharge.

L 190-191 This is certainly a valid explanation

Great to hear.

L461-463 I certainly agree with this statement

Great to hear.

L473-474 Here I also completely agree with authors' statement. Great and very timing work, badly needed for world scientific community.

Great to hear. We are happy that the need for this dataset became apparent by reading the manuscript. By adding another sentence to the introduction, we hope that you now agree that it will also become apparent at the beginning of the manuscript.

Fig. A1: The dates should be shown on these graphs

Good suggestion! We indicated the dates on the CTD profiles by coloring the lines and adding a legend.

Fig. B2: Comparison of two method of sample treatment for analyses is very useful. It is a pity that no "filtered and frozen" method was tested, because this technique is certainly the best for adequate assessment of nutrients

Please note that Figure B2 shows the comparisons for selected parameters using two protocols but the same analytical method. Dissolved inorganic nutrients as presented in Figure 12 are "filtered and frozen" following the most common protocols. We added a sentence to the Appendix B to clarify what parameters were compared:

"For a set of samples covering the period from 13 September 2019 to 2 May 2020, we measured the electrical conductivity (Fig. B1), major ion concentrations (Fig. B2), and dissolved elemental concentrations (Fig B3) measured as described in Table 1 but with two

different protocols to assess the impact of sample processing on the dissolved elemental and ion concentrations."

p.34, Fig. B3: The data on P are unclear – what do negative values mean?

This is correct, the negative concentrations have no physical significance. They reflect the detector response after calibration. Please note that all values shown here are below our detection limit indicated in Table 1 and in Figure 14. For full transparency, we decided to show these data anyways, always reporting the detection limit for potential user applications.

It is clear that for analyses of Fe, Ca, Ba, Al, on site filtration is mandatory prior to analyses. Please make sure you let it express in the text, because this is very important finding

We would like to clarify that on-site filtration for Fe, Ca, Ba, and Al was performed only for the subset of samples highlighted in green in Figure B1. For all other samples, they were frozen immediately after collection and later thawed and filtered in the lab prior to analysis. A protocol change occurred after sample #077 regarding the handling of dissolved elemental and ion concentrations (see Table 1). Specifically, samples #001 to #077 were transported unfrozen, while samples numbered >#077 were transported frozen. In Figure 14a, a noticeable offset is observed between samples <77 and >77 for Al, with the unfrozen transported samples (<77) showing lower concentrations than the frozen transported samples (>77). This mirrors the effect seen in Appendix B, where frozen and unfrozen samples from the same set were compared. Therefore, we speculate that the observed differences are more likely due to the impact of freezing versus non-freezing rather than the timing of filtration (whether performed on-site or post-transport). To definitively determine whether the filtration timing or freezing is the primary factor causing these differences, however, a split-sample approach with both processing methods would be necessary for future assessments.

We added some more description to the results:

"The different protocol (transport of samples cooled vs transport of samples frozen) between samples <#077 and >#077 resulted in visible offsets between the sample sets (i.e., F, Al, Mn, ..). The differences between unfrozen and frozen samples across different sample sets seem similar to those shown in Appendix B (comparing frozen and unfrozen samples of the same sample set)."

**Referee 2:**

General Comments:

This is a very valuable contribution. The existence and maintenance of such sustained sampling and measurement programs of physical and biogeochemical parameters of river systems is of paramount importance given the integrative nature of the information rivers carry about the corresponding watersheds, their role in linking terrestrial and marine environments and ecosystems, and their ability to reveal system-wide change. Such initiatives are of particular importance for regions of the planet that are experiencing accelerated change, such as the Arctic, where information can be used to gauge biogeochemical, and ecological responses to changing hydrological and climate conditions. Furthermore, the fate of the vast stores of carbon currently residing in permafrost in the face of on-going warming and hydrological change underlines the significance of this region in terms of global climate. With the unprecedented pace of change underway, there is the urgent need for comprehensive and intensive observation programs that provide context for this change.

Fortunately, the biogeochemistry of the major Arctic rivers have been the focus of sustained observations as a consequence of programs such as the pan-Arctic River sampling programs (PARTNERS) and Arctic Great Rivers Observatory (ArcticGRO) which extent back more than 20 years. However, these programs have been characterized by low temporal resolution, with large data gaps, particularly for specific seasons and transitional periods (shoulder seasons of freshet & freeze-up) rendering it hard to investigate different processes and constrain shorter-term variability. In such circumstances, the authors correctly highlight the limitations of models as an approach to bridge data gaps, and argue for the need for high-frequency measurements to better constrain flux estimates, and investigate short-term variability resulting from changes in hydrologic pathways and other phenomena. The articulated need for baseline observations is clear, although it is evident that marked changes are already upon us.

This present study describes a diverse suite of data acquired over a 4.5-year period from sampling at a station in the delta of the Lena River, one of the largest Arctic rivers, with a catchment dominated by permafrost. The 4+-year period covers a time interval during which winter discharge that is higher than the long-term average, and captures both record low and record high intervals of summer discharge. High-frequency (daily to weekly) sampling resulting in a total of almost 600 sampling dates, focussing exclusively on dissolved parameters. Acquisition of such detailed and long-term datasets always represents a

compromise given logistical constraints associated with ease of sampling, sampling methods and volumes, sample storage and shipment, instrumental techniques, performance and reliability, and range of parameters sought, and of course cost. Clearly, a great deal of thought and care, as well as pragmatism, has gone into the design and execution of this high-resolution sampling program. Despite some apparent limitations and inconsistencies in the dataset the existence of such high-resolution, extended datasets remains rare, and yet is of crucial value. It is not surprising for such a long-term, multi-institution and logistically challenging endeavor focused on a remote location that the datasets are somewhat heterogeneous with respect to sample processing, storage and shipment, as well as where and how the measurements were made, with some resulting patchiness in data quality. However, the manuscript benefits from a detailed description of the methods used, and discussion is provided concerning changes in methodology over the course of the observation period, which are also indicated in the figures. Analytical uncertainties in the measurements are also provided (in Table 1). For parameters where there is significant data scatter associated with measurement on specific instruments and different laboratories, and the authors caution use and interpretation of such data where this is evident (e.g., SUVA and SR in Figure 7; nutrients in Figure 11). In general, I think such discussions of data quality are satisfactory.

Thank you for your thorough and thoughtful review. We are grateful for your recognition of the importance of sustained high-frequency sampling programs, particularly in regions experiencing rapid environmental changes like the Arctic. As you noted, the integrative nature of rivers as conveyors of information from their watersheds to marine systems makes such programs crucial for understanding biogeochemical and ecological responses to climate and hydrological changes. We appreciate your acknowledgment of the logistical and methodological challenges inherent in collecting high-resolution data from remote Arctic locations, and your recognition of the value of our dataset despite these constraints. We agree that the variability in sample processing, storage, and shipment, as well as the multi-institutional nature of the program, have introduced some heterogeneity to the data. However, we have strived to be transparent in our methodology and have provided detailed discussions on the quality of the data, along with analytical uncertainties, as highlighted in your review. Your specific mention of our handling of data scatter and measurement inconsistencies in the manuscript, particularly regarding SUVA, SR, and nutrients, is greatly appreciated. We have been cautious in interpreting these datasets and have clearly indicated areas where uncertainty

may impact the results. We are pleased that you found our discussions around data quality to be satisfactory.

What was less clear is whether efforts were made to analyze splits of the same samples for the same parameters in different labs in order to address inter-lab data comparability (i.e., beyond measurement of standards). It seems that sample batches were processed in serial fashion by only one lab or another. Clarification of this point for the different parameters would be helpful. I note that in some cases comparisons were made for the same samples that were frozen versus unfrozen (Appendix B1), but what about splits of the same sample treated in the same way, but measured by different methods/research groups? One example is the water (oxygen and hydrogen) isotope data, which were obtained by mass spectrometry and optical spectroscopy. The Lena river can exhibit quite high DOM concentrations (DOC up to 20 mg L-1), which can influence spectroscopic properties. Was there any systematic comparison of water isotope data for splits of Lena water samples (not standards) between MS and CRDS methods? Irrespective, the transition in instrumental methods used in the measurement of specific parameters is indicated in the Figures, which is very helpful (e.g., water isotopes in Figure 5; DOC concentrations and absorbance in Figure 6).

Thank you for raising this important point. We acknowledge the need to clarify the efforts made to assess inter-laboratory data comparability beyond the use of standards. In this study, sample batches were primarily processed in serial fashion by different labs for different parameters, and we did not systematically analyze splits of the same sample across different labs for the same parameters. The logistical and financial effort that would be required precluded such tests. However, we did perform some comparisons for the same samples under different conditions, such as frozen versus unfrozen treatments (as shown in Appendix B1). We added a sentence to the conclusion recommending such tests in the future to improve comparability:

"Further, to improve inter-lab comparability, we recommend designated tests to measure splits of samples for the same parameters but in different labs and or using different protocols or instruments."

The data reveals some interesting contrasts for the same parameter but measured using different measurement methods (e.g., colorometric versus ion chromatographic determination of silicon concentrations; e.g., Fig. 12a) as well as different sampling handing protocols (e.g., electrical conductivity; cf. Appendix B). Such contrasts and systematic biases are to be

expected given logistical challenges in operating such a sustained measurement program. Although such offsets/biases are not optimal, the overall density of data holds promise for the potential to anticipate and correct deviations between sample suites processed and analyzed in different ways. I think these data are also highly informative for other researchers who may be applying/developing protocols for sample collection, processing and storage. Overall, I think the manuscript provides an objective assessment of the data quality and highlights key features that emerge over the time series.

Thank you very much for your review of our manuscript describing the dataset. We agree that these systematic biases, while inevitable given the logistical constraints of a sustained Arctic sampling program, do not detract from the dataset's value. In fact, as you pointed out, the density and breadth of our data allow for robust assessments and, where necessary, corrections between sample suites processed differently. We also appreciate your recognition of the dataset's potential as a resource for other researchers refining protocols for sample handling and storage in challenging environments. It is important to us to be transparent about possible limitations of the dataset that result from the unavoidable inconsistencies in sample handling and analyses. Your insights into the manuscript's objective assessment of data quality encourage us to continue refining our approaches and sharing these learnings for the benefit of the broader research community. Thank you again for your supportive comments.

Specific comments:

 - For the DOC radiocarbon data, presumably DOC concentration data is also obtained from the elemental analyzer-MICADAS AMS measurement? If so, how did DOC concentrations compare with corresponding measurements using the more conventional DOC method (high-temperature catalytic oxidation)?

DOC concentration data is not routinely obtained during radiocarbon measurements. The Elemental Analyzer used is uncalibrated; instead, $CO_2$ evolved from sample combustion can be quantified manometrically using the GIS system before injection into the AMS ion source.

- Appendix D. I am glad that the authors drew a comparison between their observations and those reported by the ArcticGRO program (albeit at a more upstream location), however, I think that this would be good to include in the main body of the manuscript as I am sure this comparison will be of direct interest to the reader. Moreover, Figure D1 and D2 clearly shows the merit of performing high temporal resolution sampling and measurement in order

to constrain (sub-)seasonal variability. It would be helpful to list which measured parameters (beyond DOC concentration and CDOM absorption) are covered by both the ArcticGRO and the present 4.5-year time series.

Thank you for this suggestion. Please see above the very similar suggestion by referee 1. We moved Appendix D to the main manuscript and added all other parameters that were measured by both programs for a comparison. We listed those parameters in the method section.

- A key question that could perhaps be addressed by the authors (at the end of the Discussion or in the Conclusions section) is whether, based on their findings, all parameters need to be measured with the same sampling frequency (given observed variability). In other words, can the data presented can be used to develop a recommended protocol for future, more streamlined sampling. For example, are there specific parameters that appear to be most diagnostic of specific (changes in) processes that are not captured in low-resolution datasets? Given the challenges (and costs) associated with sustaining such a sampling/measurement program, it might be helpful to consider things from a strategic point of view. Furthermore, would a repeat intensive phase of high-resolution sampling/measurements spanning a similar time interval be worth undertaking a decade from now? This may be particularly pertinent as I suspect maintaining this program given the current geopolitical situation will be challenging.

Specific recommendations about necessary sampling frequency for different parameters strongly depend on the question that one wants to answer and would require a tailored analysis that would be beyond the scope of this paper. We agree that this dataset might be used to answer these questions or to design sampling strategies for other sites. In the context of rapidly changing discharge, seasonality and long-term groundwater flow pathways, the goal of such monitoring must be to detect departures from predictable behavior, and emerging new relationships between observed parameters. We added a sentence to the conclusion:

"Using this dataset as a baseline, it should be the goal to repeat such sampling in the future, either as ongoing monitoring, or a repeated intense 4-year period. Future studies could utilize insights from this high-frequency sampling to determine the optimal sampling frequency needed to address specific scientific questions."

- Is any of the sampled material archived for future (repeat or new) measurements?  If so, this should be mentioned.

Good point. We added a sentence to the "Data availability" section:

"Remaining sample volumes of analyzed samples are archived at the AWI in Potsdam, Germany."